# ANGPTL8 negatively regulates NF-κB activation by facilitating selective autophagic degradation of IKKγ

Yu Zhang[1], Xian Guo[1], Wanyao Yan[1], Yan Chen[1], Mengxiang Ke[1], Cheng Cheng[1], Xiuqin Zhu[2], Weili Xue[2], Qiaoqiao Zhou[1], Ling Zheng[2], Shun Wang[3], Bin Wu[3], Xinran Liu[1], Liang Ma[1], Lianqi Huang[1] & Kun Huang[1,4]

Excessive nuclear factor-κB (NF-κB) activation mediated by tumor necrosis factor α (TNFα) plays a critical role in inflammation. Here we demonstrate that angiopoietin-like 8 (ANGPTL8) functions as a negative feedback regulator in TNFα-triggered NF-κB activation intracellularly. Inflammatory stimuli induce ANGPTL8 expression, and knockdown or knockout of ANGPTL8 potentiates TNFα-induced NF-κB activation in vitro. Mechanistically, upon TNFα stimulation, ANGPTL8 facilitates the interaction of IKKγ with p62 via forming a complex, thus promoting the selective autophagic degradation of IKKγ. Furthermore, the N-terminal domain mediated self-oligomerization of ANGPTL8 is essential for IKKγ degradation and NF-κB activation. In vivo, circulating ANGPTL8 level is high in patients diagnosed with infectious diseases, and the ANGPTL8/p62-IKKγ axis is responsive to inflammatory stimuli in the liver of LPS-injected mice. Altogether, our study suggests the ANGPTL8/p62-IKKγ axis as a negative feedback loop that regulates NF-κB activation, and extends the role of selective autophagy in fine-tuned inflammatory responses.

[1] Tongji School of Pharmacy, Huazhong University of Science & Technology, Wuhan 430030, China. [2] Hubei Key Laboratory of Cell Homeostasis, College of Life Sciences, Wuhan University, Wuhan 430072, China. [3] Department of Blood Transfusion, Wuhan Hospital of Traditional and Western Medicine, Wuhan 430022, China. [4] Centre for Biomedicine Research, Wuhan Institute of Biotechnology, Wuhan 430075, China. Correspondence and requests for materials should be addressed to K.H. (email: kunhuang2008@hotmail.com)

NF-κB plays a pivotal role in a variety of physiological and pathological processes, including inflammation, immunity and metabolism[1]. In non-stimulated cells, NF-κB is kept inactive in cytosol by binding to inhibitor of κBα (IκBα). Many agents, including pro-inflammatory cytokines, cause phosphorylation and degradation of IκBα, which results in releasing of NF-κB for translocation to the nucleus and initiating the expression of downstream genes[2]. Among these agents-mediated signaling, TNFα induction is a classical model to study the regulatory mechanisms of NF-κB activation. TNFα binds to its receptor TNF-RI to recruit the TNFR-associated death domain (TRADD), which recruits TNFR-associated factor 2 (TRAF2), TRAF5, and receptor interacting protein 1 (RIP1) to the receptor complex, then RIP1 further recruits and activates the transforming growth factor β activated kinase-1 (TAK1) complex and the IκB kinase (IKK) complex. The IKK complex consists of the catalytic subunits IKKα and IKKβ, and a regulatory subunit IKKγ. IKKγ, also known as NF-κB essential modulator (NEMO), is critical for the activation of the IKK complex; moreover, IKKγ also works as a scaffold which specifically channels the kinase activity of IKKβ to IκBα[3–5]. Finally, IκBα is phosphorylated and degraded, thereby releasing NF-κB to the nucleus.

Inappropriate NF-κB activation may cause immunodeficiency, chronic inflammation, autoimmunity and malignancy[6–8]. Therefore, IKK activation and IκBα phosphorylation, two key steps in NF-κB activation, should be tightly regulated, which highlights the physiological significance of IKKγ. Hypomorphic IKKγ mutations (IKKG gene located on the X chromosome) are lethal for male and lead to immune and developmental defects in female[9,10]; while inactivation of the negative regulators of IKKγ, such as deubiquitinase A20 and CYLD lysine 63 deubiquitinase (CYLD), leads to serious disorders[11,12]; similarly, males with a "gain of function" IKKγ mutant (ΔCT-NEMO, a C-terminal domain truncated mutant), which fails to recruit A20, develop autoinflammatory diseases[13].

Proteolysis of signaling molecules is an important way to shut down signaling transduction. The ubiquitin-proteasome system (UPS) and autophagy are two major protein degradation pathways[14]. Previously, it was thought that UPS is highly selective while autophagy is a non-selective bulk process[14]. Recent studies suggest that autophagy can also target specific protein aggregates or other substrates for degradation, referred to as selective autophagy[15,16]. In general, proteins enter the selective autophagy are first K63-polyubiquitinated, then bound by one or more of autophagy receptors such as p62, neighbor of BRCA1 gene 1 (NBR1), nuclear dot protein 52 kDa (NDP52), TOLL interacting protein (Tollip), and optineurin (OPTN), followed by engulfment in autophagosomes[14,16]. Increasing evidences suggest that autophagy is important for the inflammation and immune responses[17,18], however the role of selective autophagy in these critical physiological processes is poorly understood.

ANGPTL8 (also called Lipasin, RIFL, TD26 or C19orf80) is known as a key regulator of plasma lipid metabolism, which functions mainly by inhibiting lipoprotein lipase[19,20]. Here, we demonstrate intracellular ANGPTL8 as a novel negative feedback regulator of TNFα-mediated NF-κB activation, which may work as a critical step to avoid excessive inflammatory responses by facilitating p62-mediated autophagic IKKγ degradation.

## Results

### Pro-inflammatory cytokines up-regulate ANGPTL8.
ANGPTL8 regulates lipid metabolism, and the level of circulating ANGPTL8 is increased in type 2 diabetes (T2D)[21,22]. Since lipid toxicity and T2D are tightly correlated with inflammation, we investigated the level of ANGPTL8 upon stimulation of pro-inflammatory

cytokines such as TNFα and IL-1β. In HepG2 cells, the transcription and expression of ANGPTL8 were both significantly increased after being treated with TNFα (Fig. 1a, b), with a TNFα dose-dependent elevation in ANGPTL8 level (Fig. 1c). Similar results were observed in two additional cell lines, HEK293T (a human embryotic kidney cell line) and A549 (a human lung cancer cell line) (Supplementary Fig. 1a), although their ANGPTL8 level was markedly lower than that of HepG2 cells (Supplementary Fig. 1b). Consistently, IL-1β treatment induced the transcription and expression of ANGPTL8 in HepG2 cells (Supplementary Fig. 1c, d). Collectively, these results indicate that the ANGPTL8 expression could be triggered by different inflammatory stimuli and in various cells.

### Knockdown or knockout of ANGPTL8 potentiates NF-κB activation.
We next determined whether ANGPTL8 regulates NF-κB activation. Three ANGPTL8-RNAi plasmids were generated, which efficiently inhibited the transcription and expression of ANGPTL8 in HepG2 cells (Fig. 2a). In luciferase reporter assays, knockdown of ANGPTL8 enhanced TNFα- or IL-1β-induced NF-κB activation (Fig. 2b and Supplementary Fig. 2a), and the level of NF-κB activation was correlated with the knockdown efficiency. As a negative control experiment, ANGPTL8-RNAi did not affect the IFNγ-induced IRF1 activation, which is another pathway involved in immunity and separated from TNFα-mediated signaling (Fig. 2c). These data indicate that ANGPTL8 specifically potentiates the TNFα-triggered and IL-1β-triggered NF-κB activation.

Further quantitative real-time PCR (qPCR) analysis demonstrated that knockdown of ANGPTL8 significantly potentiated the TNFα-triggered transcription of NF-κB downstream genes, such as IL8, CXCL2, and NFKBIA (Fig. 2d). ANGPTL8-RNAi-#3 was used in following experiments for its highest efficiency. Similar to HepG2 cells, knockdown of ANGPTL8 also potentiated the TNFα- or IL-1β-triggered NF-κB activation in HEK293T and A549 cells (Supplementary Fig. 2b, c). Consistently, knockdown of ANGPTL8 enhanced TNFα-induced phosphorylation of IKKs and IκBα, two hallmarks of NF-κB activation (Fig. 2e).

To confirm the role of ANGPTL8 in TNFα-mediated NF-κB activation, we generated ANGPTL8-deficient HepG2 cell lines by using the CRISPR-Cas9 system. The ANGPTL8-deficient ($ANGPTL8^{-/-}$) clones were confirmed at DNA and protein levels (Fig. 3a, b). In reporter assays, the $ANGPTL8^{-/-}$ cells showed significantly enhanced NF-κB activation comparing to the wild-type cells after TNFα or IL-1β induction (Fig. 3c and

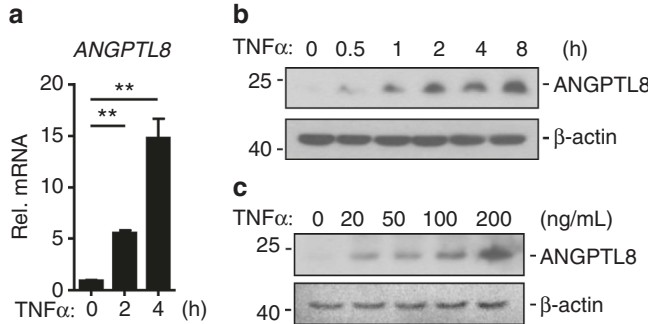

**Fig. 1** TNFα upregulates the expression of ANGPTL8 in HepG2 cells. **a**, **b** The transcription level (**a**, $n = 3$) and protein level (**b**) of ANGPTL8 after TNFα (50 ng/mL) treatment. The following experiments used the same dose of TNFα if not mentioned. **c** The protein level of ANGPTL8 at 8 h after different dosages of TNFα. Data are shown as the mean ± SEM, unpaired two-tailed student's test was used for statistics (**a**), $**p < 0.01$. Data are representative of three independent experiments

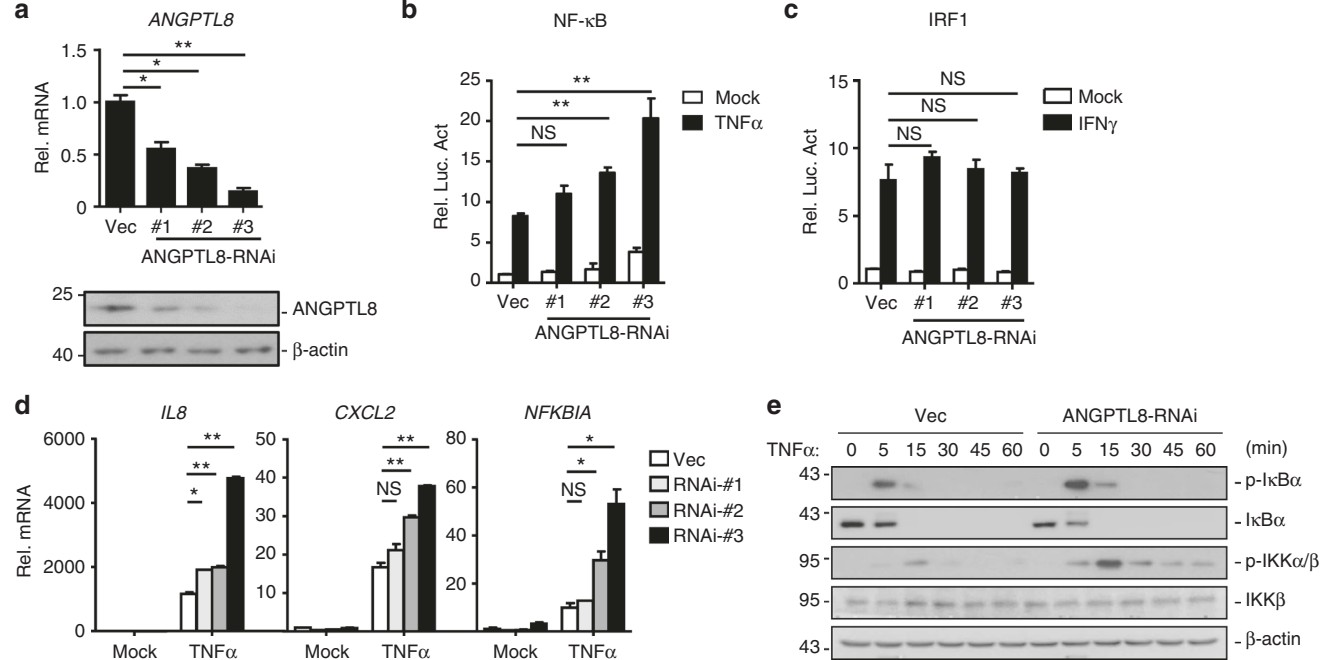

**Fig. 2** Knockdown of ANGPTL8 potentiates TNFα-induced NF-κB activation. **a** Efficacy of three different ANGPTL8-RNAi plasmids on the transcription (upper panels) and expression (lower panels) of ANGPTL8 in the control or shANGPTL8 stable HepG2 cell lines ($n = 3$). **b** Effects of ANGPTL8-RNAi on TNFα-induced NF-κB activation in the cell lines indicated as in **a** ($n = 3$). **c** Effects of ANGPTL8-RNAi on IFNγ (100 ng/mL) induced IRF1 activation ($n = 3$). **d** Effects of ANGPTL8-RNAi on TNFα-induced IL8, CXCL2 and NFKBIA transcription for 2 h in cell lines indicated as in **a** ($n = 3$). **e** Effects of ANGPTL8-RNAi on TNFα-induced IKK and IκBα phosphorylation in ANGPTL8-RNAi-#3 or control cells in **a**. Data are shown as the mean ± SEM, unpaired two-tailed Student's test was used for statistics (**a**–**d**), *$p < 0.05$, **$p < 0.01$, NS > 0.05. Data are representative of three independent experiments

Supplementary Fig. 2d). Knockout of ANGPTL8 facilitated TNFα-induced *IL8*, *CXCL2* and *TNFA* transcription (Fig. 3d). In contrast, the level of IFNγ-induced *STAT1* transcription was comparable between the wild-type and *ANGPTL8*$^{-/-}$ cells (Fig. 3e). We next reconstituted the ANGPTL8 knockout cells with Flag-tagged ANGPTL8 by retrovirus-mediated gene transfer (Fig. 3f), and found that reconstitution of ANGPTL8 into *ANGPTL8*$^{-/-}$ cells suppressed the TNFα-triggered *IL8*, *CXCL2* and *TNFA* transcriptions (Fig. 3g). Collectively, we demonstrate that endogenous ANGPTL8 negatively regulates NF-κB signaling.

**ANGPTL8 regulates NF-κB activation at the IKK complex level.** We next investigated the molecular mechanisms by which ANGPTL8 regulates the TNFα or IL-1β signaling mediated NF-κB activation. TNFα/IL-1β-mediated NF-κB activation includes three major steps: adaptors such as TRAF2/6-mediated or RIP1-mediated recruitment of IKK complex to TNFR, activation of IKKβ, and activation of NF-κB. We cotransfected HEK293T cells with a vector expressing TRAF2, TRAF6, RIP1, IKKβ, or p65, together with an overexpression or knockdown vector of ANGPTL8 plus an NF-κB luciferase reporter vector. The over-expression/knockdown of ANGPTL8 induced changes of NF-κB activation upon overexpression of TRAF2/6 or RIP1, but not IKKβ and p65 (Fig. 4a, b). These data implicate that ANGPTL8 may involve in the recruitment or activation of IKK complex.

Consistently, by co-IP experiments, we found that ANGPTL8 interacted with RIP1, IKKβ and IKKγ in the cells co-overexpressed ANGPTL8 and the regulator molecules involved in NF-κB activation cascade (Fig. 4c). Moreover, in untransfected cells, ANGPTL8 rapidly interacted with IKKβ/γ and RIP1 upon TNFα treatment (Fig. 4d), which may in part due to the TNFα-triggered ANGPTL8 expression. Collectively, these results

indicate that ANGPTL8 may target IKK complex through participating in the recruitment or activation of IKKs.

Since ANGPTL8 is mostly known as a secreted protein, after we found its role in the regulation of intracellular signaling, we studied the localization of ANGPTL8. In ANGPTL8 over-expression HepG2 cells, a large amount of ANGPTL8 was detected in cell lysate, while a proportion of ANGPTL8 was secreted (Supplementary Fig. 3a), this result was consistent with a recent study carried out in HEK293T cells[23]. Immunofluorescence experiments confirmed that endogenous ANGPTL8 had intracellular localization (Supplementary Fig. 3b).

**ANGPTL8 facilitates the degradation of IKKγ.** We next investigated how ANGPTL8 regulates the IKK complex. Interestingly, we found that overexpression of ANGPTL8 resulted in markedly decreased expression level of Flag-IKKγ, but not Flag-IKKα, -IKKβ, -TRAF6, or -RIP1 (Fig. 5a), whereas knockdown of ANGPTL8 showed the opposite effects (Fig. 5b). Furthermore, knockdown of ANGPTL8 attenuated TNFα-induced degradation of IKKγ but not that of IKKα/β (Fig. 5c), without affecting the transcription of IKKγ (Supplementary Fig. 4). These data suggested that ANGPTL8 promotes the degradation of IKKγ. Besides, knockdown or knockout of ANGPTL8 significantly potentiated the TNFα-induced transcription of IL8 and CXCL2, which was abolished by further knockdown of IKKγ (Fig. 5d, e), suggesting that ANGPTL8 facilitates TNFα-induced NF-κB activation by degrading IKKγ.

**ANGPTL8 mediates autophagic IKKγ degradation.** Protein degradation is one of the key ways to turn off signaling transduction. Proteins in eukaryotes can be degraded by UPS or autophagy. ANGPTL8-mediated IKKγ degradation was completely blocked by 3-methyladenine (3MA) and chloroquine (CQ), inhibitors for autophagosome and lysosome, respectively;

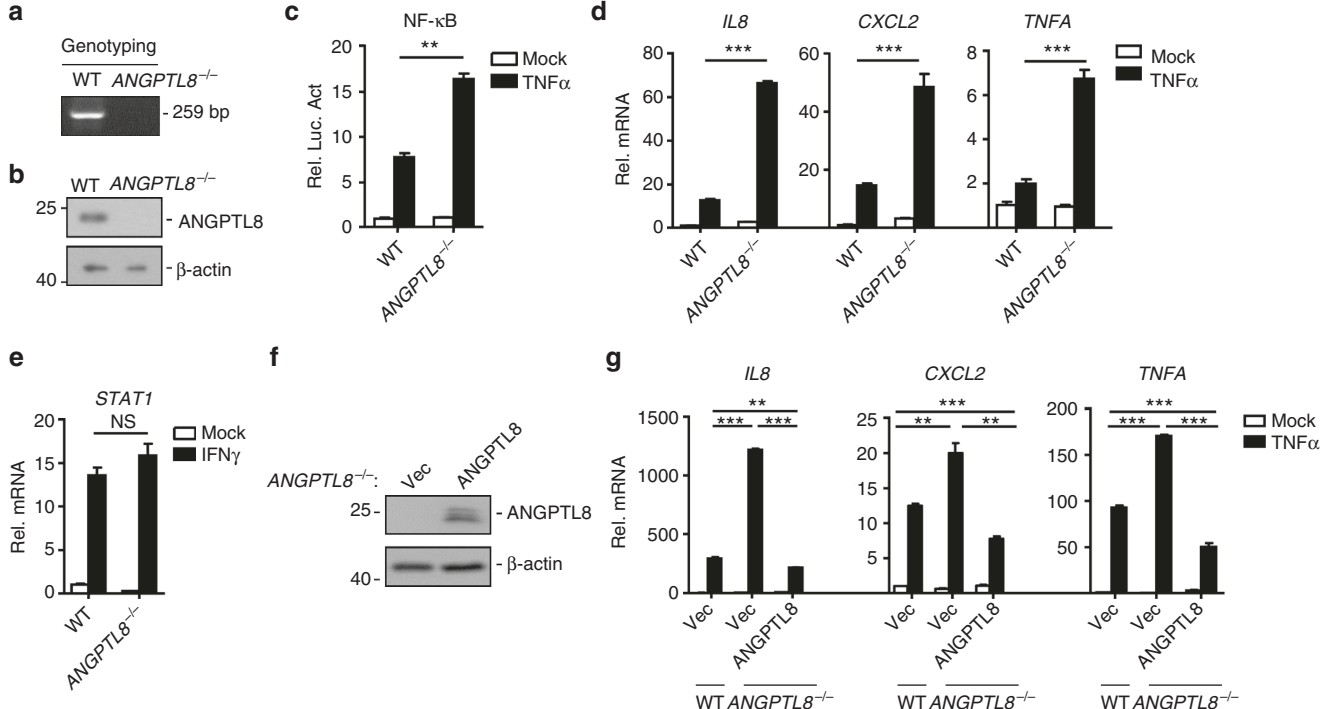

**Fig. 3** Knockout of ANGPTL8 potentiates TNFα-induced NF-κB activation. **a** Genotyping of *ANGPTL8*[−/−] and wild-type HepG2 cells. **b** The expression level of ANGPTL8 in *ANGPTL8*[−/−] and wild-type HepG2 cells. **c** Effects of ANGPTL8 deficiency on TNFα-induced NF-κB activation in luciferase reporter assays (*n* = 3). **d** Effects of ANGPTL8 deficiency on TNFα-induced *IL8*, *CXCL2* and *TNFA* transcription (*n* = 3). **e** Effects of ANGPTL8 deficiency on IFNγ-induced *STAT1* transcription (*n* = 3). **f** Reconstitution of ANGPTL8-Flag in the *ANGPTL8*[−/−] cell lines. **g** Effects of ANGPTL8 deficiency and reconstitution on TNFα-induced *IL8*, *CXCL2*, and *TNFA* transcription (*n* = 3). Data are shown as the mean ± SEM, unpaired two-tailed student's test was used for statistics (**c**, **d**, **e**, **g**), **p* < 0.01, ***p* < 0.001, NS > 0.05. Data are representative of three independent experiments

whereas the proteasome inhibitor MG132 could not inhibit the ANGPTL8-mediated IKKγ degradation (Fig. 6a). Autophagy-related 5 (ATG5) and ATG7 are essential adaptors for the autophagic degradation, and knockdown of ATG5/7 inhibited the rapamycin-induced turnover of LC3 (Supplementary Fig. 5)[24,25]. Consistently, ANGPTL8-mediated IKKγ degradation was dramatically attenuated in ATG5-/ATG7-RNAi cells (Fig. 6b). These results suggest ANGPTL8 mediates IKKγ degradation in a macroautophagy-dependent manner.

Confocal microscopy experiments further suggested that in unstimulated cells, only a small fraction of IKKγ was co-localized with GFP-LC3 dots, a marker for autophagosome; whereas in cells overexpressing ANGPTL8, the overlap and correlation efficiency between IKKγ and GFP-LC3 dots were markedly increased (Fig. 6c, d). It has been observed that TNFα stimulation induces relocalization of IKKγ into punctate structures that are enriched in activated IKK kinases and IKKγ, which may be essential for NF-κB activation[26,27]. Consistently, we found that TNFα stimulation induces the recruitment of IKKγ into punctate structures, however, this TNFα-induced IKKγ punctual relocalization was dramatically decreased in *ANGPTL8*[−/−] cells, and the co-localization of IKKγ with GFP-LC3 was also significantly reduced (Fig. 6e, f).

**ANGPTL8 and p62 co-mediate the autophagic degradation of IKKγ.** Motivated by the observations that ANGPTL8 selectively mediated the autophagic degradation of IKKγ but not IKKα/β, we next studied whether ANGPTL8 mediates IKKγ degradation via selective autophagy. K63-linked ubiquitin chains have been reported to promote selective autophagy-dependent degradation for specific target proteins[14], we found overexpression of wild-type or K63-linked ubiquitin (the ubiquitin in which all of the

lysine mutated to arginine except the lysine 63) enhanced the ANGPTL8-mediated IKKγ degradation (Fig. 7a and Supplementary Fig. 6); whereas CYLD, which removes the K63-linked ubiquitin chains of IKKγ[28], attenuated ANGPTL8-mediated IKKγ degradation in a dose-dependent manner (Fig. 7b).

Typically, cargoes are selectively transferred by one or more of the known autophagy receptors (p62, Tollip, NDP52, NBR1 and OPTN etc.) for degradation[15,29]. We found that it was p62 and Tollip, but not NDP52, NBR1 or OPTN, that mediated the degradation of IKKγ, which was further markedly enhanced when co-expressed with ANGPTL8 (Fig. 7c and Supplementary Fig. 7a). In wild-type cells, although p62 and Tollip both significantly and dose-dependently induced IKKγ degradation, ANGPTL8 deficiency only reversed p62-mediated, but not Tollip-mediated degradation of IKKγ (Fig. 7d and Supplementary Fig. 7b). Moreover, ANGPTL8 could interact with p62, but not Tollip; and the interaction between ANGPTL8 and p62 was greatly enhanced after TNFα treatment (Fig. 7e). Also, knockdown of p62 dramatically attenuated ANGPTL8-mediated IKKγ degradation (Fig. 7f). The LC3 interacting region (LIR) of p62 is responsible for recruitment of LC3 to the autophagosome[30]. Here, we observed that p62ΔLIR (a LIR domain deleted p62 mutant) showed dramatically decreased ability in mediating IKKγ degradation (Supplementary Fig. 7c). Notably, similar to p62 and IKKγ, we found that ANGPTL8 also underwent autophagic degradation (Supplementary Fig. 7d). Altogether, these data indicate that ANGPTL8 and p62 work collaboratively to mediate the autophagic degradation of IKKγ.

**Reducing ANGPTL8 attenuates the IKKγ–p62 interaction.** In selective autophagy, substrate recognition by autophagy receptor is essential for cargo selection. By performing co-IP experiments

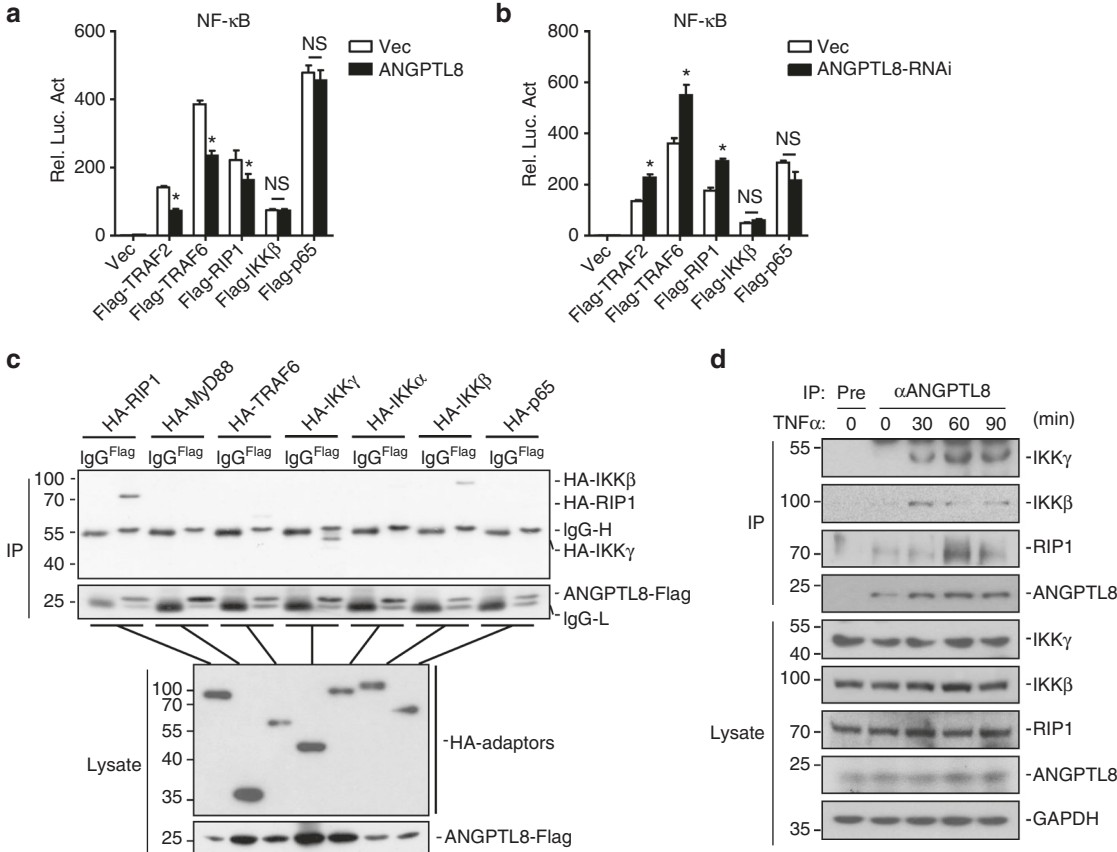

**Fig. 4** ANGPTL8 regulates TNFα-induced NF-κB activation at the IKK complex level. **a** Effects of the ANGPTL8 overexpression on Flag-TRAF2-, Flag-TRAF6-, Flag-RIP1-, Flag-IKKβ- or Flag-p65-mediated NF-κB activation. HEK293T cells were transfected with a control vector or ANGPTL8-Flag expression plasmid (0.05 μg) and the indicated plasmids (0.1 μg of TRAF2 or IKKβ, 0.01 μg of TRAF6, RIP1 or p65). Reporter assays were performed 20 h after transfection (n = 3). **b** Effects of the ANGPTL8-RNAi on Flag-TRAF2-, Flag-TRAF6-, Flag-RIP1-, Flag-IKKβ- or Flag-p65-mediated NF-κB activation. HEK293T cells were transfected with a control vector or ANGPTL8-RNAi-#3 plasmid (0.3 μg) for 24 h before being transfected with the indicated plasmids. Twenty hours later, the cells were lysed and reporter assays were performed (n = 3). **c** ANGPTL8 interacts with IKKγ, IKKβ and RIP1 in the mammalian overexpression system. **d** Endogenous ANGPTL8 interacts with IKKγ, IKKβ and RIP1 in HepG2 cells. Data are shown as the mean ± SEM, unpaired two-tailed student's test was used for statistics (**a**, **b**), *p < 0.05, NS > 0.05. Data are representative of three independent experiments

in physiological conditions, we found that IKKγ, ANGPTL8 and p62 interacted with each other (Fig. 8a, b). We noted that only a small fraction of p62 was associated with IKKγ in resting cells, and this association was greatly increased after TNFα treatment, which is consistent with the observation that ANGPTL8 was simultaneously recruited to IKKγ (Fig. 8a, b). Consistently, TNFα-induced interaction between IKKγ and p62 was impaired by knockdown or knockout of ANGPTL8 (Fig. 8c, d). These data indicate that upon TNFα stimulation, IKKγ, p62, and ANGPTL8 form a complex in which ANGPTL8 plays an important role in mediating the interaction.

**Oligomerization of ANGPTL8 is essential for IKKγ degradation.** In selective autophagy, protein aggregation is a key event in mediating cargo selection and separation[31]. Interestingly, by co-IP analysis, we found ANGPTL8 can self-oligomerize (Fig. 9a). ANGPTL8 is predicted to have an N-terminal signal peptide (residues 1-25) and two coiled-coil (CC) domains (residues 77–134 and 156–193) which are presumably associated with protein–protein interaction (Fig. 9b); however, the function of different domains/regions in ANGPTL8 is not clear[32,33]. By using domain mapping, we identified the region between residues 26–70 as an essential domain for the self-oligomerization of ANGPTL8 (Fig. 9a). Notably, the truncation mutants (71-C and

Δ26–70) that lost the self-association capacity could not mediate the IKKγ degradation, while the CC domain that unrelated to the self-oligomerization of ANGPTL8 was dispensable in this process (Fig. 9c and Supplementary Fig. 8). Consistently, ANGPTL8-Δ26–70 showed a significantly diminished interaction with IKKγ and lost the ability of inhibiting TNFα-induced NF-κB activation (Fig. 9d, e). Thus, oligomerization of ANGPTL8 mediated by its N-terminal 26–70 region is essential for the ANGPTL8-mediated inhibition of NF-κB activation by mediating the interaction and degradation of IKKγ. Moreover, we found that the coiled-coil domains were responsible for the ANGPTL8-p62 interaction (Supplementary Fig. 9). Interestingly, compared with the full-length ANGPTL8, ANGPTL8-Δ26–70 showed a stronger interaction with p62, suggesting the self-oligomerization is not required for ANGPTL8-p62 interaction (Fig. 9d).

Gel filtration analysis was performed to explore the oligomerization status of ANGPTL8. The full-length ANGPTL8 was most abundant in a higher molecular mass (fractions 5–12) than ANGPTL8-Δ26–70 (fractions 10–15) (Fig. 9f). Transfection of ANGPTL8-Δ26–70 led to significantly decreased molecular weight of p62 and IKKγ-containing fractions in the same experiments, indicating the importance of ANGPTL8 oligomerization in the formation of IKKγ and p62-containing protein complex. We also expressed and purified the recombinant ANGPTL8 to assess its oligomerization tendency in vitro

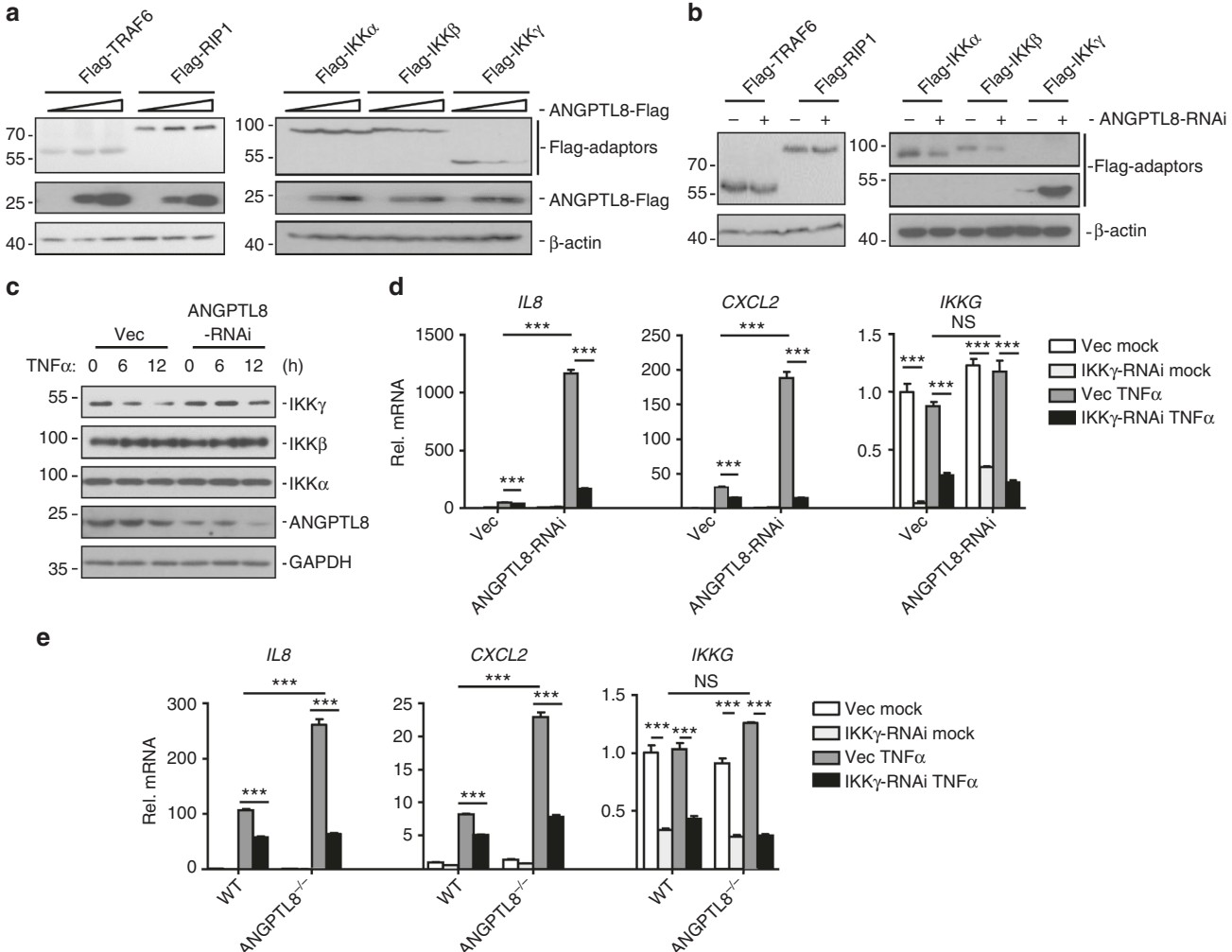

**Fig. 5** ANGPTL8 facilitates the degradation of IKKγ. **a** Effects of ANGPTL8 overexpression on the expression level of transfected TRAF6, RIP1, IKKα, IKKβ or IKKγ in HEK293T cells. **b** Effects of ANGPTL8-RNAi on the expression level of transfected TRAF6, RIP1, IKKα, IKKβ, or IKKγ. **c** Effects of ANGPTL8-RNAi on the expression levels of the endogenous IKKα, IKKβ or IKKγ after TNFα and CHX co-treatment. The indicated cells were treated by CHX (200 μg/mL) for 1 h, followed by TNFα treatment for the indicated time. **d** Knockdown of IKKγ inhibits ANGPTL8-RNAi facilitated NF-κB activation triggered by TNFα (2 h) in the ANGPTL8-RNAi or control cell lines (n = 3). **e** Knockdown of IKKγ inhibits ANGPTL8 deficiency-facilitated NF-κB activation triggered by TNFα for 2 h (n = 3). Data are shown as the mean ± SEM, unpaired two-tailed student's test was used for statistics (**d**, **e**). ***$p < 0.001$, NS > 0.05. Data are representative of three independent experiments

(Supplementary Fig. 10a). The circular dichroism spectra revealed that in solution, ANGPTL8 adapted a mixed α-helix/β-structures/random coil structure (Supplementary Fig. 10b, c), which agrees with the structural prediction (Supplementary Fig. 10d). ANGPTL8 showed tendency to aggregate into oligomers and large aggregates (fibrils) as measured by the dynamic light scattering (DLS) assays or by probing with the anti-oligomer and anti-fibril antibodies (Supplementary Fig. 10e, f). Under transmission electronic microscopy, oligomers and fibrils were also identified (Supplementary Fig. 10g).

**Potential involvement of ANGPTL8 in acute inflammation.** Our in vitro studies suggest that ANGPTL8 degrades IKKγ, whose tight regulation is essential for the balance of inflammation, we next investigated the physiological relevance of ANGPTL8 in vivo. As demonstrated by qPCR analysis (Supplementary Fig. 11A), *Angptl8* is relatively high in the liver and brown adipose tissue (BAT), and low in the spleen, lung and kidney of mouse. Next, we examined the tissue level of *Angptl8* in

mice challenged with lipopolysaccharide (LPS), a constituent of the Gram-negative bacteria outer membranes and an important microbial trigger that stimulates innate immunity. Interestingly, in tissues with high level of *Angptl8*, fast upregulation and downregulation of *Tnfa* transcription during the acute phase (0–1 h) and the resolution phase (1–6 h) was respectively observed upon LPS challenge; in contrast, they were relatively slow in tissues with low *Angptl8* expression (Fig. 10a and Supplementary Fig. 11b). This observation implicates that for tissues that are sensitive to the inflammatory stress, larger amount of "brake" molecules, such as ANGPTL8, may be demanded. Furthermore, upon LPS stress, the expression of *Angptl8* was increased, while *Ikkγ* expression was decreased (Fig. 10b, c); however, the interaction among *Ikkγ*, p62 and *Angptl8* was enhanced (Fig. 10d). These results are consistent with our in vitro experiments.

Third, we measured the circulating ANGPTL8 level in the blood of two groups of patients with systemic inflammatory response syndrome. One group includes patients with positive detection of procalcitonin (PCT) which is a biomarker for early

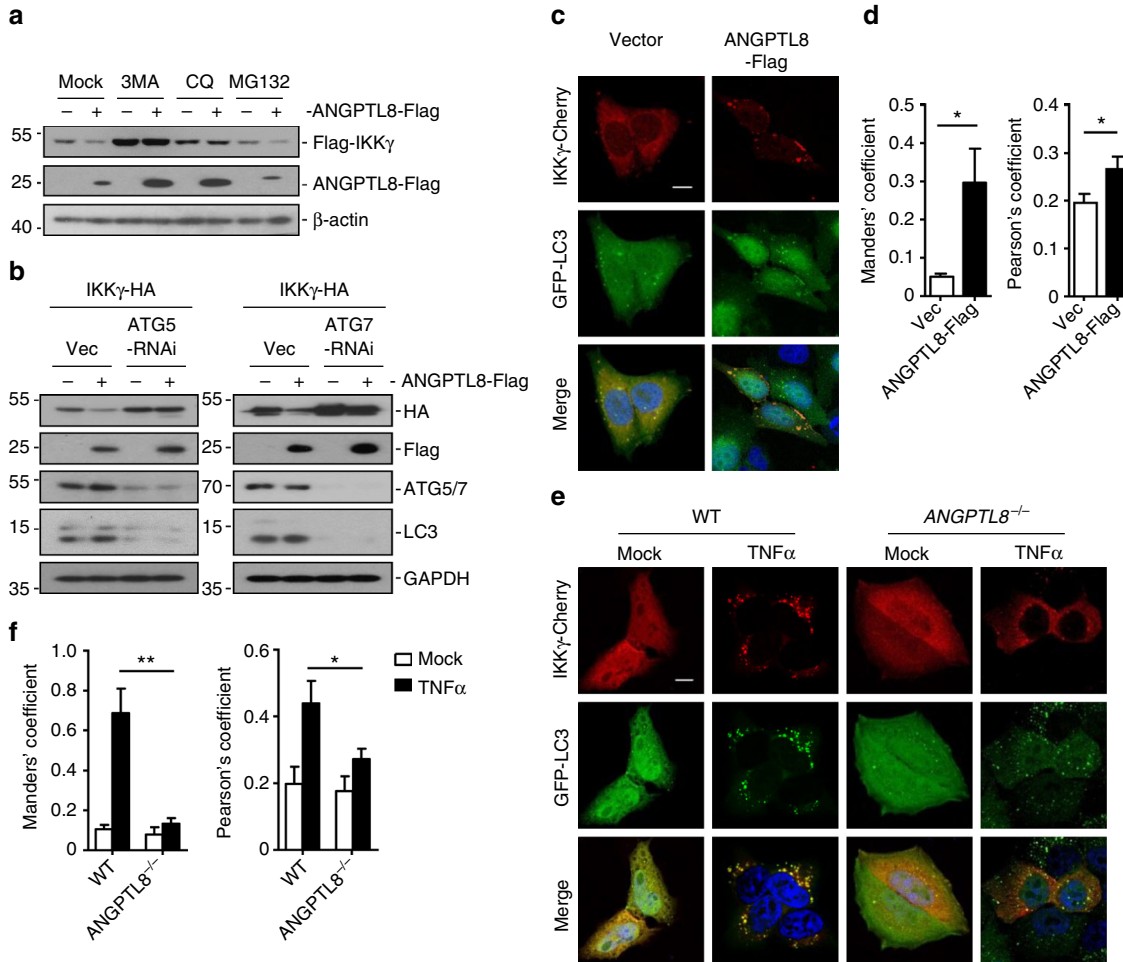

**Fig. 6** ANGPTL8 mediates the autophagic IKKγ degradation. **a** Effects of 3MA, CQ, and MG132 on ANGPTL8-mediated destabilization of IKKγ. HEK293T cells ($1 \times 10^5$) were transfected with the indicated plasmids. Fourteen hours after transfection, the cells were treated with the indicated inhibitors (3MA, 3.35 mM; CQ, 0.05 mM; MG132, 10 μM) for 6 h before immunoblots. **b** Effects of the knockdown of ATG5/7 on ANGPTL8-mediated IKKγ proteolysis. The HEK293T was transfected with ATG5/7-RNAi for 12 h and followed by puromycin selection (1 μg/mL) for 12 h before the indicated plasmids were transfected for 24 h, and then cells were collected for immunoblots. **c, d** The representative images (**c**) of and quantitative results (**d**) of effects of ANGPTL8 on the translocation of IKKγ to autophagosomes, the scale bar represents 10 μm ($n = 6$). **e, f** The representative images (**e**) and quantitative results (**f**) of effects of ANGPTL8 deficiency on TNFα-induced co-localization of IKKγ with the autophagosomes. The scale bar represents 10 μm ($n = 6$). Data are shown as the mean ± SEM, unpaired two-tailed student's test was used for statistics (**d, f**). *$p < 0.05$, **$p < 0.001$. Data are representative of three independent experiments

diagnosis of sepsis[27], the other group includes patients with positive detection of endotoxin which is an important microbiological assessment for Gram-negative bacteria-mediated inflammation[34]. Compared to the healthy subjects, the circulating ANGPTL8 level was dramatically increased in both groups of patients (Fig. 10e and Supplementary Table 1). Collectively, these results indicated that ANGPTL8 can be induced by inflammatory stimuli in mouse and human, and may thus play roles in shutting down acute inflammatory response.

## Discussion
ANGPTL8 has been known as a potent regulator of lipid metabolism[35]. Circulating ANGPTL8 is increased in patients with T2D or non-alcoholic fatty liver diseases, which makes ANGPTL8 an attractive therapeutic target for metabolic syndromes[22,36,37]. On the other hand, inflammation, especially the pro-inflammatory cytokines-mediated chronic inflammation, has been demonstrated to contribute to the development of metabolic diseases such as T2D. However, the role of ANGPTL8 in

inflammation is unknown. As a central event of inflammation and immunity, TNFα-induced NF-κB activation must be tightly controlled to avoid inflammatory diseases, autoimmunity and cancers[6–8]. Here we present the first evidence that multiple inflammatory stimuli, including TNFα, induce the transcription and expression of ANGPTL8 in vitro and in vivo; the latter then forms a protein complex with p62 and IKKγ, in which ANGPTL8 works as a co-receptor of p62 and facilitates the autophagic IKKγ degradation, thereby inhibiting the TNFα-induced NF-κB activation (working model shown in Fig. 10f). The ANGPTL8/p62-IKKγ axis thus serves as a negative feedback loop to restrict the TNFα-trigged NF-κB activation and inflammation.

Since ANGPTL8 is mostly known as a secreted protein, it is interesting that our results suggest it also plays intracellular roles and has intracellular location. Actually, other studies have demonstrated that some secreted proteins have intracellular functions. For example, ISG15, an interferon-induced modifier, is found both intracellularly and extracellularly; the secreted ISG15 has cytokine like activities[38]; whereas the intracellular ISG15 can conjugate various proteins via ISGylation, and prevents the IFN-

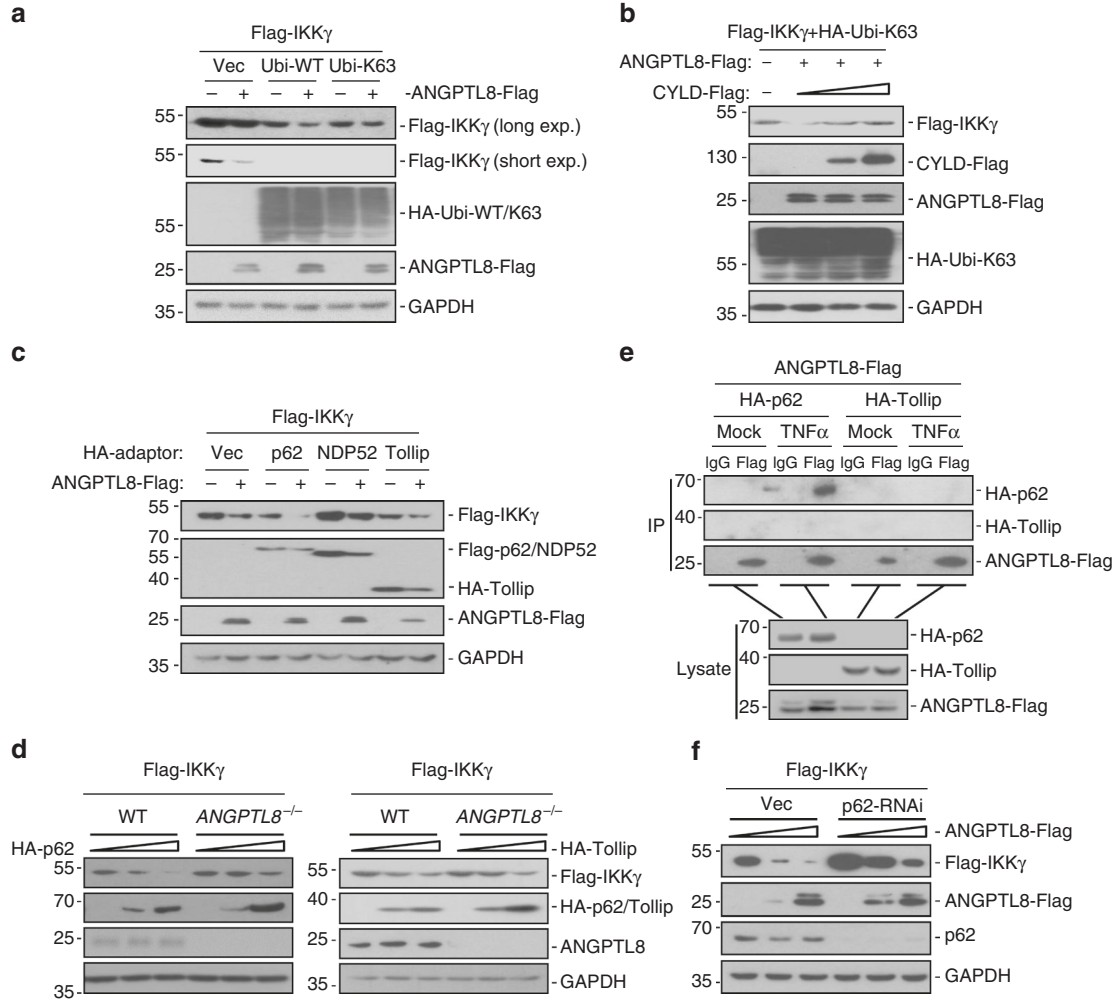

**Fig. 7** ANGPTL8 and p62 collaboratively mediate the autophagic degradation of IKKγ. **a** The effects of ubiquitin on the ANGPTL8-mediated IKKγ degradation in HEK293T cells. Ubi-WT, the wild-type ubiquitin; Ubi-K63, the ubiquitin in which all of lysine (K) mutated to arginine (R) except K63. **b** The effect of deubiquitinase CYLD on the ANGPTL8-mediated IKKγ degradation. **c** The effects of different autophagic adaptors on the ANGPTL8-mediated IKKγ degradation. **d** Overexpression of p62 but not Tollip-mediated IKKγ degradation is attenuated in *ANGPTL8⁻/⁻* cells. **e** ANGPTL8 interacts with p62 but not Tollip in the mammalian overexpression system. **f** ANGPTL8-mediated IKKγ degradation is decreased in p62-RNAi stable cell line. Data are representative of three independent experiments

α/β-dependent autoinflammation[39]. PCSK9 is another secreted protein, and binds hepatotic LDLRs both extracellularly and intracellularly which leads to LDLRs degradation[40,41]. Similarly, the intracellular function of ANGPTL8 has been reported. It can enhance the cleavage of ANGPTL3, a molecule involved in the triglyceride metabolism[20]; and it is also involved in the lipolysis of adipocytes[42].

ANGPTL8 has been reported to be induced by thyroid hormone, and modulates autophagy mainly via enhancing autolysosome maturation[43]. In this study, we found that ANGPTL8 specifically mediates the autophagic IKKγ degradation, which can be promoted by overexpression of K63-linked ubiquitin, and chemical or genetic blockage of autophagy could inhibit this IKKγ degradation (Figs. 5–7). Furthermore, p62, a classical autophagy receptor, works cooperatively with ANGPTL8 to mediate the proteolysis of IKKγ (Figs. 7 and 8 and Supplementary Fig. 7). Although it has been reported that IKKγ is mainly degraded by lysosomal pathway[44], this is the first report that IKKγ can be degraded by ANGPTL8/p62-mediated selective autophagy.

The specificity of selective autophagy for the degraded cargoes is mainly attributed to autophagy receptors. While there are numerous molecules or cell organelles need to be degraded, only a handful of known autophagy receptors are responsible for this process[15,31,45]. There is a general model behind the selectivity: the substrate needs to be recognized by other proteins (e.g., molecular chaperones) before interacting with autophagy receptors, which can be termed as "co-receptor"[31]. In this study, several lines of evidence indicate ANGPTL8 as a co-receptor of p62 to mediate the IKKγ selection. First, p62 and many other chaperones are stress-responsive, possibly to ensure different substrates are properly degraded under certain conditions[46,47]; similarly, ANGPTL8 is a stress-responsive molecule with enhanced expression under TNFα, IL-1β, LPS, or infection (Figs. 1 and 10 and Supplementary Fig. 1). Second, the interactions between p62 and ANGPTL8 are mutually required for IKKγ degradation (Fig. 7). Third, TNFα treatment promotes the formation of the IKKγ-p62-ANGPTL8 complex, which is impaired in the ANGPTL8 knockdown or knockout cells (Fig. 8). Consistently, enhanced binding affinity between Ikkγ, Angptl8, and p62 in the liver of LPS-treated mice was also observed (Fig. 10d). Our results thus suggested a novel role of ANGPTL8 as a co-receptor of p62 in selective autophagy.

After cargo selection, an important role for autophagy receptors such as p62 is to sequester cargoes into larger aggregates

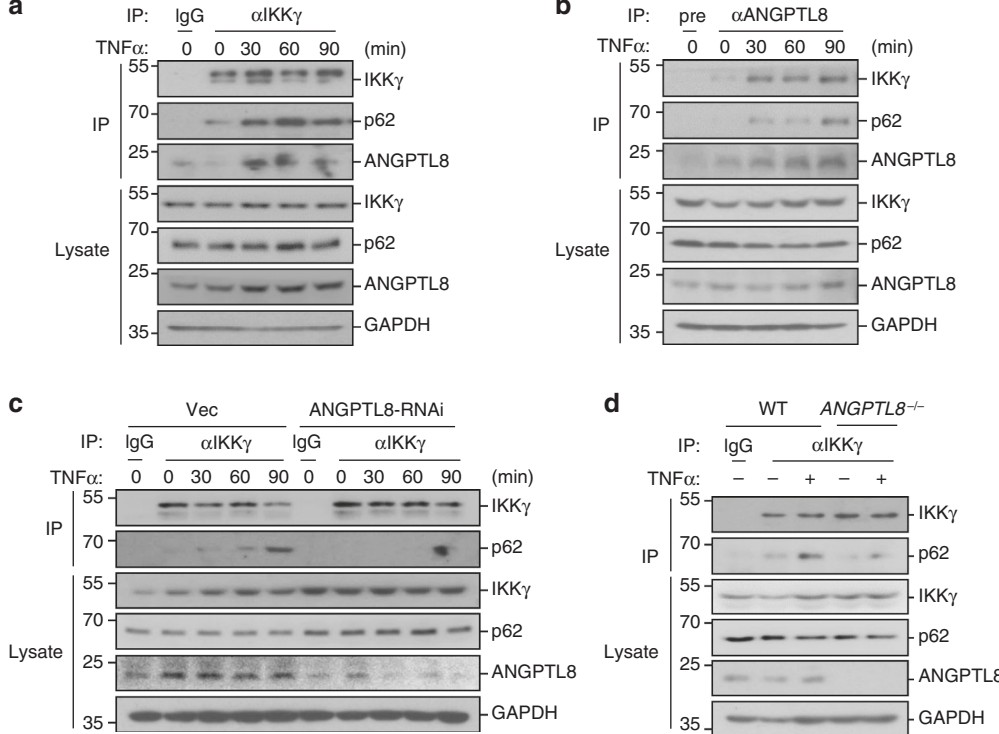

**Fig. 8** ANGPTL8 is important for the IKKγ-p62 interaction. **a, b** ANGPTL8 forms a complex with IKKγ and p62 upon TNFα stimulation. HepG2 cells were stimulated with TNFα or left unstimulated for indicated time ($2.5 \times 10^7$ cells per group). And then cells were lysed and cell lysate was subjected to co-IP analysis by anti-IKKγ. (**a**) or anti-ANGPTL8 (**b**) followed immunoblots. **c** The effects of ANGPTL8-RNAi on the IKKγ-related protein complex formation. The experiment was performed as in **a** except the ANGPTL8-RNAi or control cells ($2.5 \times 10^7$) were used. **d** The effects of ANGPTL8 deficiency on the IKKγ-related protein complex formation. ANGPTL8-deficient or control cells ($2.5 \times 10^7$) were left untreated or treated with TNFα for 1 h before subjected to Co-IP analysis and immunoblots. Data are representative of three independent experiments

before they being degraded[31,48,49]. During this process, p62 forms oligomers, and such suprastructure enables its interaction with LC3B and ubiquitinated cargoes[49,50]. It is an interesting and open question that whether other p62-related proteins work in similar ways[50]. Here, we demonstrate that ANGPTL8 self-oligomerizes through its N-terminal region and such oligomerization is essential for the interaction/degradation of IKKγ, thereby affecting the ANGPTL8-mediated inhibition of TNFα-induced NF-κB activation (Fig. 9), indicating the oligomerization is important to the interaction between the co-receptors and cargoes. However, ANGPTL8 seems to directly interact with p62, since ANGPTL8-Δ26–70 showed stronger interaction with p62 compared with the full-length ANGPTL8 (Fig. 9).

As major protein degradation pathways, UPS and autophagy have been proven to play significant roles in inflammation[17,18]. While there have been dozens of regulators identified to influence NF-κB activation by UPS, reports on how autophagy degrades specific signaling molecules in inflammation are rare[50–53], which may in part due to the concept that UPS is highly selective but autophagy is a bulk process. However, recent studies suggested both process can be specific, UPS is involved in the rapid degradation of single proteins, while autophagy can selectively remove protein aggregates and damaged/excess organelles that are too big in size for proteasomes[14]. Interestingly, signaling components, especially receptors and scaffold proteins, tend to form oligomers to mediate the signaling[54], implicating that these aggregation-prone proteins may be the appropriate substrates for autophagy. IKKγ is a well-known scaffold protein and TNFα stimulation can induce the IKKγ translocation to supramolecular structures which is important to the NF-κB activation[4,5,27,55]. Here, we demonstrated that IKKγ could undergo the ANGPTL8/

p62-facilitated selective autophagy, we also observed that the percentage of punctual IKKγ can be enhanced by ANGPTL8, these findings suggested a possibility that aggregated IKKγ is not only important for signaling transduction, but also a precondition for its degradation. Moreover, our study implicates that additional scaffold proteins in this or other pathways may also be degraded with similar mechanism.

Multiple studies have confirmed the induction of circulating ANGPTL8 in different metabolic syndromes[36,37]. Here we found that the level of circulating ANGPTL8 was dramatically increased in severe infection (Fig. 10e). To our knowledge, it is the first report on the relationship between ANGPTL8 and acute inflammation in clinical patients. We also observed a correlation of Angptl8 level with LPS-induced acute inflammatory response in different tissues of mouse (Fig. 10a and Supplementary Fig. 11). Future experiments using Angptl8 knockout or transgenic mice will be helpful to reveal the mechanisms underlying its physiological and pathological roles in inflammatory diseases.

Collectively, our results uncover an important fine-regulation mechanism for NF-κB activation. Notably, the observations that ANGPTL8 also can be induced by additional factors such as IL-1β, LPS, insulin resistance and feeding from this study and from literature[56], implicate a more broaden role of ANGPTL8 in the autophagic degradation of other inflammation or metabolism associated proteins, a question awaits further exploration.

## Methods

**Reagents, antibodies and cell lines**. Immunoblots with ANGPTL8/Angptl8 were done by mouse anti-ANGPTL8 monoclonal antibody (1:300), a kind gift from Dr. Y. Wang (Wuhan University)[19,20]. Immunofluorescence with ANGPTL8 was done by an anti-ANGPTL8 monoclonal antibody (Sigma, SAB3501080, 1:100). Co-IP

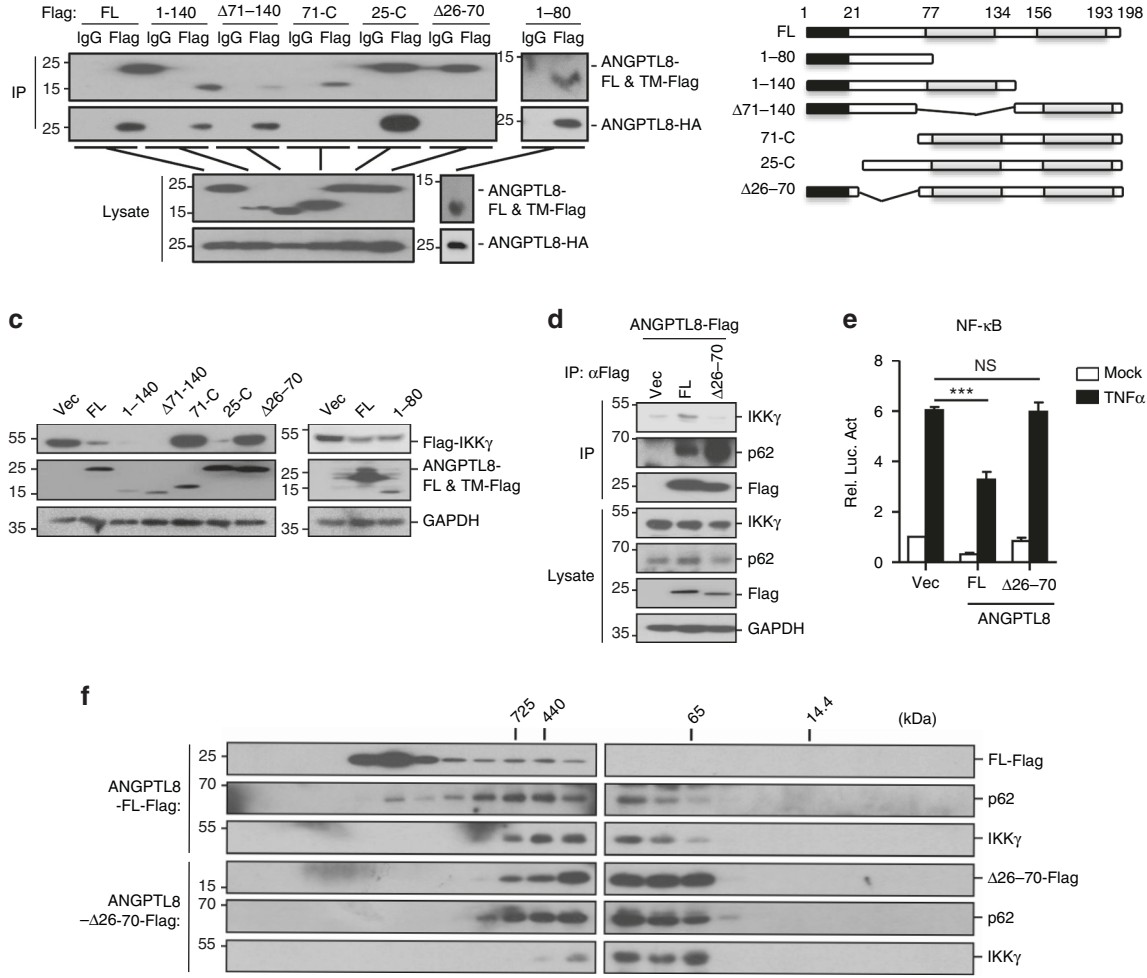

**Fig. 9** ANGPTL8 self-aggregation is essential for the degradation of IKKγ. **a** Domain mapping of the ANGPTL8 self-aggregation in HEK293T cells. **b** A schematic presentation of human ANGPTL8 and its truncation mutants. **c** The effects of ANGPTL8 and its truncation mutants on the degradation of IKKγ in HEK293T cells. **d** The ANGPTL8-Δ26–70 shows a diminished interaction with IKKγ. Three dishes of HEK293T cells ($2 \times 10^6$ cells per dish) were transfected with indicated plasmids, 24 h later, cells were lysed and subjected to co-IP and followed immunoblots by indicated antibodies. **e** The effects of the full-length or −Δ26−70 truncation mutant of ANGPTL8 on the TNFα-induced NF-κB activation ($n = 3$). **f** Analysis of protein complex containing ANGPTL8, p62 and IKKγ by size-exclusion chromatography. HEK293T cells ($2 \times 10^6$) were transfected with indicated plasmids for 24 h later before cells were lysed and subjected to size-exclusion chromatography. Data are shown as the mean ± SEM in **e**, unpaired two-tailed Student's test was used for statistics (**e**). ***$p < 0.0001$, NS > 0.05. Data are representative of three independent experiments. FL full length, TM truncation mutants

experiments with ANGPTL8/Angptl8 were done by mouse anti-ANGPTL8 poly-clonal antibody which was raised against recombinant human full-length ANGPTL8 with standard protocols. A list of commercial reagents, and other antibodies and dilutions used in the present study was provided in Supplementary Table 2. HepG2 (CL-0103000), HEK293T (CL-0005), and A549 (CL-0016) cells, which were analyzed with authenticated STR locus and tested for mycoplasma contamination, were purchased from Procell Biotech. (Wuhan, China).

**Constructs.** NF-κB, IRF1 and TK luciferase reporter plasmids, mammalian expression plasmids for Flag-tagged RIP1, MyD88, TRAF6, IKKα, IKKβ, IKKγ and p65; HA-tagged Ubi (WT, K63), prepared as previously described[57,58], were kind gifts of Dr. H.-B. Shu (Wuhan University). EGFP tagged OPTN were purchased from Addgene (#27052). CYLD-Flag is a kind gift of Dr. B. Zhong (Wuhan University). Lentiviral GFP-LC3 is a kind gift from Dr. Z.Y. Song (Wuhan University); Flag- or HA-tagged IKKγ; ANGPTL8 and their truncated mutants; HA-tagged p62, NDP52, and Tollip; Flag-tagged NBR1, p62, p62ΔLIR and Cherry-tagged IKKγ were constructed with standard procedures.

**Transfection and luciferase reporter gene assays.** Cells ($5 \times 10^4$) were seeded on 48-well plates and transfected on the following day, empty control plasmid was added to ensure that each transfection receives the same amount of total DNA. To normalize transfection efficiency, 0.02 μg of pRL-TK Renilla luciferase reporter

plasmid was added to each transfection. Luciferase assays were performed using a dual-specific luciferase assay kit (Promega), the firefly luciferase activities (NF-κB or IRF1 firefly luciferase reporter) were normalized based on Renilla luciferase activities.

**RNAi experiments.** Double-strand oligonucleotides corresponding to the target sequences were cloned into the pSuper plasmids (Oligoengine). The target sequences for human ANGPTL8, p62, IKKγ, ATG5, and ATG7 cDNA are listed in Supplementary Table 3.

**Retrovirus-mediated stable RNAi cell lines.** The packaging cell line HEK293T was transfected with retroviral vectors by calcium phosphate precipitation. Twelve hours later, cells were washed by PBS, and antibiotics-free medium was added for another 24 h. The filtered supernatant was used to infect HepG2 or ANGPTL8$^{-/-}$ cells in the presence of 6 μg/mL polybrene. The infection was repeated twice to enhance transduction efficiency.

**Quantitative real-time PCR.** Total RNA was isolated from cells using RNAiso Plus reagent (Takara) and subjected to qPCR analysis. The mRNA levels of specific genes were normalized to GAPDH. The gene-specific primer sequences for ANGPTL8, IKKG, Angptl8, and Tnfa are listed in Supplementary Table 3. The

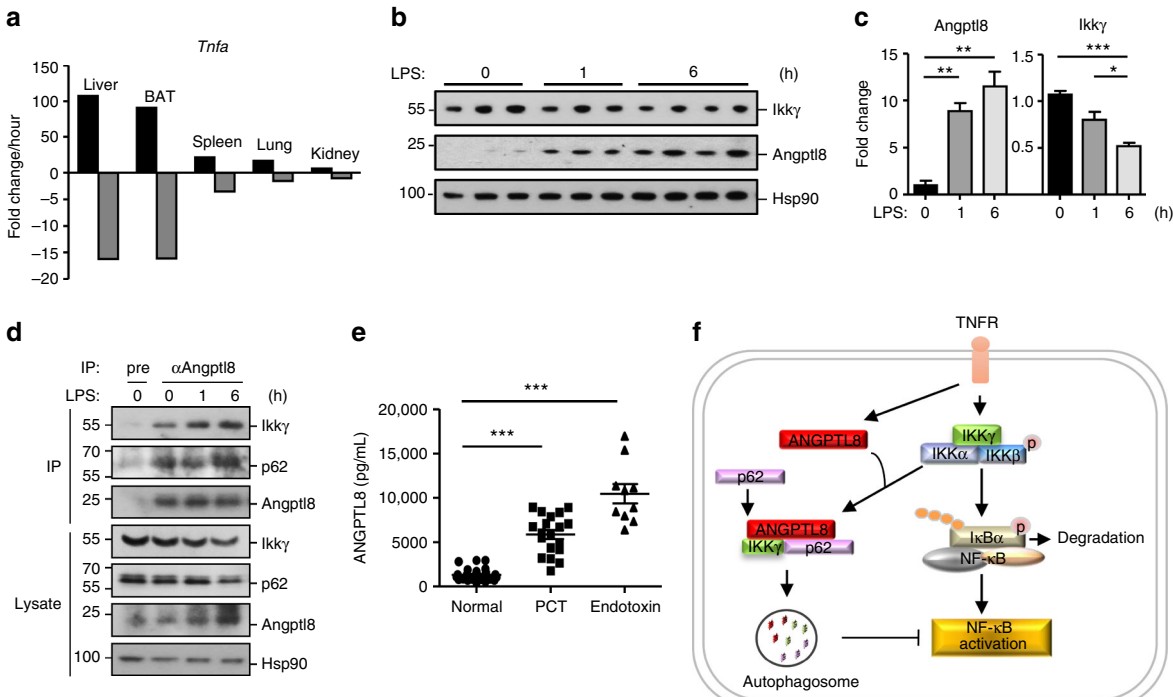

**Fig. 10** Potential involvement of ANGPTL8 in the acute inflammation caused by infection. **a** The upregulation and downregulation rate of *Tnfa* transcription during the acute phase (0–1 h) and the resolution phase (1–6 h) upon LPS infection. The upregulation rate was determined by the fold change between 0–1 h, while the downregulation rate was determined by the fold change per hour between 1–6 h after the injection. (*n* = 3 for 0 and 1 h, *n* = 4 for 6 h). **b**, **c** The immunoblots (**b**) and quantitation results (**c**) of the expression of Ikkγ and Angptl8 in the liver upon LPS infection. Each lane stands for one mouse. **d** Angptl8 forms a complex with Ikkγ and p62 upon LPS infection. Equal amount of lysate obtained from each animal at the same time point were mixed, and subjected to Co-IP experiment (*n* = 3 for 0 and 1 h, *n* = 4 for 6 h). **e** The circulating ANGPTL8 was elevated in patients with positive detection of PCT or endotoxin (healthy control, *n* = 30; PCT, *n* = 18; endotoxin, *n* = 10). **f** A working model of ANGPTL8 mediated regulation of TNFα-induced signaling. TNFα upregulates the expression of ANGPTL8, and then aggregated ANGPTL8 facilitates the ANGPTL8-p62-IKKγ complex formation, which promotes the autophagic IKKγ degradation and inhibits TNFα-induced NF-κB activation. Data are shown as the mean ± SEM in **c**, **e**, unpaired two-tailed Student's test was used for statistics. ***p < 0.0001, **p < 0.01, *p < 0.05

primers for *CXCL2*, *IL8*, *NFKBIA*, *TNFA*, and *GAPDH* were as we previously described[58,59].

**CRISPR-Cas9-mediated knockout of ANGPTL8.** The CRISPR-Cas9 based protocols for genome engineering were used as described[60]. PGL-U6-GRNA and PST1374-Cas9 plasmids were gifts of Dr. X. Zhang (Wuhan University). The ANGPTL8 gRNA target sequence and the identification primers for ANGPTL8 knockout are listed in Supplementary Table 3.

**Confocal microscopy.** The WT and *ANGPTL8*−/− stable cell lines expressing GFP-LC3 were constructed by lentivirus mediated gene transfer. At 20 h after transfected with IKKγ-Cherry, cells were treated with or without TNFα for 2 h. After fixing with 4% (W/V) formaldehyde, the nuclei were stained by DAPI (1 μg/mL) and the cells were imaged with a Zeiss LSM 880 confocal microscope. The plugin JACoP of Image J was used to calculate the colocalization rate for the red pixels and green dots (green dots represent LC3-II, a marker for autophagosome[61]) as described[62]. Colocalization of signals from IKKγ-Cherry and GFP-LC3 dots was evaluated using Manders' overlap coefficient and the Pearson's correlation coefficient.

**Coimmunopreicipitation assays.** Transfected HEK293T cells (~5 × 10⁶) or HepG2 cells (~2 × 10⁷) were lyzed in l mL pre-lysis buffer (20 mM Tris-HCl, pH 7.4, 150 mM NaCl, 1 mM EDTA, 1% Triton X-100, 10 μg/mL aprotinin, 10 μg/mL leupeptin, 0.5 mM β-glycerophosphate disodium salt hydrate and 1 mM phenylmethylsulfonyl fluoride). For each immunoprecipitation, 0.8 mL of cell lysate was incubated with 0.5 μg of the indicated antibody and 35 μL of 50% slurry of GammaBind Plus-Sepharose (Amersham Biosciences) at 4 °C for 4 h. The Sepharose beads were then washed three times with 1 mL of lysis buffer containing 500 mM NaCl. The precipitates were resuspended by 60 μL SDS loading buffer, and subsequent immunoblot analysis was performed with indicated antibodies.

**Size-exclusion chromatography.** Cells (2 × 10⁶) were transfected with indicated plasmids for 24 h before being lysed in 500 μL of pre-lysis buffer. The cell lysate was then incubated on ice for 30 min followed by sonication and was spun down at

12000×g for 10 min. The supernatant was filtered with a 0.45 μm filter (Millipore) before being loaded onto a Superose 6 size-exclusion chromatography column (GE Healthcare, 1 × 30 cm), which was pre-equilibrated with Triton and EDTA free pre-lysis buffer. The samples were eluted at 4 °C by lysis buffer at a flow rate of 500 μL/min and collected in fractions of 500 μL. The fractions were precipitated with 20% trichloroacetic acid and analyzed by immunoblots with antibodies against Flag, IKKγ, and p62.

**Human studies.** To compare circulating ANGPTL8 levels between patients with inflammation and healthy controls, patients with positive detection of procalcitonin (PCT > 0.5 μg/L, *n* = 18), positive detection of endotoxin (LPS > 0.1 EU/mL, *n* = 10) and healthy controls (*n* = 30, from physical examination center) were included in the study. The detail sample information was listed in Supplementary Table 1, Circulating levels of human ANGPTL8 were determined by enzyme immunoassay kit (EIAab Science, Wuhan, China). Informed consent was obtained from all subjects and the Ethical approval ((2017)09) was obtained by the Medical Ethics Committee of the Wuhan Hospital of Traditional and Western Medicine (Wuhan First Hospital).

**Mice.** Male C57BL/6 mice were obtained from the Center for Animal Experiment/ Animal Biosafety Level-III Laboratory of Wuhan University. Mice were housed in ventilated microisolator cages with free access to water and normal chow. Animals were handled according to the Guidelines of the China Animal Welfare Legislation, as approved by the Committee on Ethics in the Care and Use of Laboratory Animals of College of Life Sciences, Wuhan University. For LPS injection experiment, two-month-old mice were randomly divided into three groups and intraperitoneally (i.p.) injected with a single dose of LPS (3 mg/kg) for 1 or 6 h, untreated age- and sex-matched littermate as controls (*n* = 3 for 0 and 1 h, *n* = 4 for 6 h). The mice were killed, liver, brown fat, spleen, lung, and kidney were collected.

**Far-UV circular dichroism and structural modeling.** A JASCO-810 circular dichroism spectropolarimeter (Tokyo, Japan) was used to monitor secondary structures. Incubated samples were diluted to a final concentration of 10 μM and detected in a 1 mm path length at 25 °C. Circular dichroism (CD) spectra were

obtained from 260 to 200 nm at a 200 nm/min scanning speed and a 2 nm bandwidth. All samples were measured in triplicates and the averages were taken. The data were converted to mean residue ellipticity and the secondary structural contents were further calculated with the software CDPro[63]. Structural modeling of ANGPTL8 was conducted using the online server I-TASSER.

**Dynamic light scattering analysis**. The sizes of ANGPTL8 aggregates were measured by dynamic light scattering in a zeta pals potential analyzer (Brookhaven Instruments, USA). Samples were vortexed and detected at room temperature, the scattering angle was set at 90°. Each measurement was repeated three times and the average mean particle size was recorded. The data was analyzed by the multimodal size distribution (MSD) software[64].

**Dot blot assays**. Sample aliquotes (2 μL) obtained at indicated time points were blotted onto a nitrocellulose membrane (Bio-Rad, USA). Dried membrane was blocked with 5% non-fat milk for 1 h at room temperature and then incubated with anti-oligomer antibody (A-11) or anti-fibril antibody (OC) at 4 °C overnight. The membrane was incubated with anti-rabbit IgG for 2 h at room temperature later. An ECL chemiluminescence kit (Advansta, USA) was used for the development.

**Transmission electronic microscopy**. Incubated solution was applied onto a 300-mesh formvar-carbon coated copper grid and sit for 5 min. Freshly prepared uranyl formate (1%) was dropwise added for staining. Dried samples were observed under a transmission microscope (Hitachi, Japan) operating at an accelerating voltage of 200 kV[65].

**Statistical analysis**. Sample sizes, as described in figure legends, were selected based on effect size and availability as per usual standard. Randomization was done by selecting animals of similar age and weight. No blinding was involved in animal studies. Statistically significant differences between the mean values were determined by two-tailed Student's $t$-test ($*p < 0.05$, $**p < 0.01$, $***p < 0.001$, NS $> 0.05$). Data are presented as the mean $\pm$ SEM.

**Data availability**. The data that support the findings of this study is provided in the supplementary information (Supplementary Figs. 12–22) or available upon request.

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

## Acknowledgements

We thank Drs. H.-B. Shu, B. Zhong, C.-Q. Lei, M.-M. Hu, H. Chen, and W.-J. Wang for technical help and stimulating discussions. This study was supported by the Natural Science Foundation of China (31500706, 31471208, and 31671195), Natural Science Foundation of Hubei Province (2015CFB245), the Fundamental Research Fund for the Central Universities (2016YXMS143), the Integrated Innovative Team for Major Human Diseases Program of Tongji Medical College, and the Front Youth Program of HUST.

## Author contributions

Y.Z. and K.H. designed research; Y.Z., X.G., W.-Y.Y., Y.C., M.-X.K., C.C., X.-Q.Z., W.-L.X., Q.-Q. Z., L.M., and L.-Q.H. performed research; B.W., S.W., X.-R.L., and L.Z. contributed new reagents/analytic tools; Y.Z., X.G., W.-Y.Y., Y.C., C.C., L.Z., and K.H. analyzed data; and Y.Z., L.Z., and K.H. wrote the paper.

## Additional information

**Competing interests:** The authors declare no competing financial interests.

