## [Peer Review File · Nature Communications]

Reviewers' comments:

Reviewer #1 (Remarks to the Author):

This submission describes an interaction between ANGPTL8 and the p62-IKK γ signaling pathway to negatively regulate NF- κ B activation induced by TNF α . ANGPTL8 is a secreted protein which modulates lipoprotein lipase-mediated plasma triglyceride clearance. p62 is an important intracellular autophagy receptor, which can complex with IKK γ to promote degradation of the latter.

The authors provide mechanistic details how ANGPTL8 facilitates the interaction of IKK γ with p62. However, they do not address the key question how a secreted protein with signal sequence and with well-established function in the extracellular space can get to the cytosol to exert its interactions with p62-IKK γ . This is a major flaw of the manuscript and questions the overall findings of the study.

Another problem is that their findings are restricted overexpression/RNAi/CRIPR-cas9 in cell lines and lack of any in-vivo supporting data, as well as proper controls that would show these effects in NF- κ B activation are not due to stress (in overexpression experiments) or off-target effects (in RNAi experiments).

Specific comments:

Fig. 1D shows ANGPTL8 expression in kidney HEK293 cells and lung A549 cells. This is somewhat a surprising finding since ANGPTL8 is not expressed in kidney or lung or in HEK293 and A549 cells according to the protein atlas database. What are the controls that ANGPTL8 is being detected? It is not clear if the experiments are from a single study or replicated in individual experiments.

The paper claims that ANGPTL8 regulates NF- κ B at the IKK complex level. This is shown in Figure 4, where overexpression of IKK β and p65 shows no effect in the presence of overexpression of ANGPTL8 (Fig 4B). However, authors do not explain why they do not see activation by IKK β and p65 overexpression when ANGPTL8 is knockdown (Fig 4B).

Western blot for RIP1 in Fig 4D should be replaced with better quality image.

Authors claim that ANGPTL8 has a specific effect on the degradation of IKK γ as shown in Fig 5 A. However, this seems not to repeat in experiment in Fig 7A (Vec control). Authors claim that the degradation of IKK γ effect is specific for HA-p62. However, in Fig 7D HA-Tollip also seems to have an effect in IKK γ degradation. Quantitation of the Western blots might clarify these effects.

Reviewer #2 (Remarks to the Author):

In this manuscript, Zhang et al. showed that ANGPTL8 and p62/SQSTM1 negatively regulate NF- κ B signaling through autophagic degradation of IKK γ . The paper is well written and the data well presented, but contained some issues to solve before publication in Nature Communications.

Comments

1. In Figure 6A, to indicate the autophagy-dependent IKK γ degradation, the authors used pharmacological inhibitors such as 3-methyladenine and chloroquine. But, those reagents have side effects other than inhibition of autophagy. The authors should carry out the experiments with Atg-knockout cells (e.g., Atg5, Atg7 and FIP200-knockout cells).
2. In Figure 6B, the authors should use cells that stably express GFP-LC3 at low level since LC3 is an aggregate-prone protein (Kuma et al., *Autophagy*. 2007 Jul-Aug;3(4):323-8.).

3. In Figure 9, the authors identified a region of ANGPTL8, which is needed for its self-oligomerization and the interaction with IKK γ . Using an ANGPTL8 mutant defective IKK γ -interaction, the authors should examine whether the mutant lacks an ability to repress NF- κ B activation in response to TNF α -stimulation.
4. The authors should test whether autophagic degradation of IKK γ is impaired by expression of a coiled-coil domain mutant of ANGPTL8.
5. The authors may investigate a necessity of LC3-interacting region (LIR) of p62/SQSTM1 in the IKK γ degradation.
6. In IP-IB analysis shown in Figure 4D, the blot for RIP1 was not so clear. Replace it with the representative blot.

Reviewer #3 (Remarks to the Author):

The manuscript from Huang and colleagues describes the role of a protein involved in lipid metabolism Angiopoietin-like protein 8 (ANGPTL8) in the regulation of the autophagic degradation of IKK γ under TNF α activation. The authors show that TNF α induces the expression of ANGPTL8 which reduces the activation of NF κ B in TRAF2/TRAF6/RIP1-dependent pathway, suggesting that ANGPTL8 targets the IKK complex. Then they show the ANGPTL8-mediated autophagic degradation of IKK γ using overexpression and knockdown of ANGPTL8. The authors suggest that this occurs through homo- and hetero-dimerization of ANGPTL8 with IKK γ and p62. The study includes a detailed characterization of the mechanism by which ANGPTL8 interacts with IKK γ and p62.

It's an interesting topic and the molecular mechanisms proposed are novel, although the physiological relevance and impact is not addressed. In general the experiments are robust and interpretation appropriate, however, some issues should be addressed before the study could be considered for publication:

- 1) Fig5C; the authors claim only IKK γ is a target for ANGPTL8 mediated autophagy, but levels of IKK α also seem to decrease after ANGPTL8 knockdown.
- 2) Fig6; stable cell lines expressing GFP-LC3 should be used since transient transfection of GFP-LC3 can lead to formation of autophagy-independent dots due to high overexpression (Kuma et al., 2007).
- 3) Fig6C; how has co-localization dots % has been measured? Pearson's and Mander's coefficient should be used (Bolte and Cordelieres, 2006).
- 4) Fig6D; a clear redistribution of IKK γ from a diffuse localisation to vesicles appears under TNF α stimulation, that does not occur in the absence of ANGPTL8. Is this TNF-induced oligomerisation of IKK γ upon TNF-signalling? This observation supports the data of the authors but is not discussed.

Bolte, S., and F.P. Cordelieres. 2006. A guided tour into subcellular colocalization analysis in light microscopy. *J Microsc.* 224:213-32.

Kuma, A., M. Matsui, and N. Mizushima. 2007. LC3, an autophagosome marker, can be incorporated into protein aggregates independent of autophagy: caution in the interpretation of LC3 localization. *Autophagy.* 3:323-8.

Point-by-Point Response Letter

Re: Manuscript NCOMMS-17-01143 entitled “ANGPTL8 negatively regulates NF- κ B activation by facilitating selective autophagic degradation of IKK γ ” by Yu Zhang *et al.*

We would like to thank the reviewers and the editor for the comments and insightful suggestions on our manuscript, which have greatly helped us to improve our study. Based on these comments, we have performed new experiments and clarified certain statements in our revised manuscript. The revised places are marked in blue fonts in the revision.

Since the experimental process of construction/purchase, breeding and experiments on the physiological relevance of *Angptl8* by using knockout or transgenic mice may need years, we instead investigated some physiological related phenomena about ANGPTL8 in a LPS-injected mouse model as well as in blood samples of clinical diagnosed infection diseases. These new results clearly indicated that the level of ANGPTL8 is responsive to inflammatory stimuli in both mouse and human, and thus may participate in the control of the inflammatory responses (please see below for detail). The following is our point-by-point response to the reviewers' comments.

Reviewer #1 (Remarks to the Author):

This submission describes an interaction between ANGPTL8 and the p62-IKK γ signaling pathway to negatively regulate NF- κ B activation induced by TNF α . ANGPTL8 is a secreted

protein which modulates lipoprotein lipase-mediated plasma triglyceride clearance. p62 is an important intracellular autophagy receptor, which can complex with IKKg to promote degradation of the latter.

The authors provide mechanistic details how ANGPTL8 facilitates the interaction of IKKg with p62. However, they do not address the key question how a secreted protein with signal sequence and with well-established function in the extracellular space can get to the cytosol to exert its interactions with p62-IKKg. This is a major flaw of the manuscript and questions the overall findings of the study.

Reply:

We agree with the reviewer that ANGPTL8 is widely known as a secreted protein. We performed new experiments to identify the localization of ANGPTL8 in cultured cells. As shown in revised supplementary Figure 3a, after overexpression of ANGPTL8 in HepG2 cells, a large amount of ANGPTL8 was detected in cell lysate, while a proportion of ANGPTL8 was secreted. This result was consistent with a recent study carried out in HEK293T cells¹. In addition, immunofluorescence experiments also confirmed that the endogenous ANGPTL8 has intracellular localization (revised supplementary Fig. 3b). Collectively, these results suggested that ANGPTL8 can be detected intracellularly, where it plays a role in the proteolysis of IKK γ as we proposed in the manuscript.

It has been reported that some secreted proteins show intracellular functions. For example, ISG15, an interferon-induced modifier, is found both intracellularly and extracellularly; the secreted ISG15 has cytokine like activities²; whereas the intracellular ISG15 can conjugate various proteins via ISGylation, and also prevent the IFN- α/β -dependent autoinflammation³. PCSK9 is another secreted protein, which binds LDLRs both extracellularly and intracellularly and leads to LDLRs degradation in hepatocytes^{4,5}. The intracellular functions

of ANGPTL8 also have been reported. It can enhance the cleavage of ANGPTL3, a molecule involved in the triglyceride metabolism⁶; and it is also involved in the intracellular lipolysis of adipocytes⁷. We discuss this question at page 16-17 of the revised manuscript.

1. Xun Chi et. al. ANGPTL8 promotes the ability of ANGPTL3 to bind and inhibit lipoprotein lipase. *Molecular Metabolism*, (2017). <http://dx.doi.org/10.1016/j.molmet.2017.06.014>
2. Bogunovic D, et al. Mycobacterial disease and impaired IFN-gamma immunity in humans with inherited ISG15 deficiency. *Science* **337**, 1684-1688 (2012).
3. Zhang X, et al. Human intracellular ISG15 prevents interferon-alpha/beta over-amplification and auto-inflammation. *Nature* **517**, 89-93 (2015).
4. Lagace TA, et al. Secreted PCSK9 decreases the number of LDL receptors in hepatocytes and in livers of parabiotic mice. *The Journal of clinical investigation* **116**, 2995-3005 (2006).
5. Poirier S, et al. Dissection of the endogenous cellular pathways of PCSK9-induced low density lipoprotein receptor degradation: evidence for an intracellular route. *The Journal of biological chemistry* **284**, 28856-28864 (2009).
6. Quagliarini F, et al. Atypical angiopoietin-like protein that regulates ANGPTL3. *Proceedings of the National Academy of Sciences of the United States of America* **109**, 19751-19756 (2012).
7. Mysore R, Liebisch G, Zhou Y, Olkkonen VM, Nidhina Haridas PA. Angiopoietin-like 8 (Angptl8) controls adipocyte lipolysis and phospholipid composition. *Chemistry and physics of lipids*, (2017).

Another problem is that their findings are restricted overexpression/RNAi/CRIPR-cas9 in cell lines and lack of any in-vivo supporting data, as well as proper controls that would show these effects in NF-kb activation are not due to stress (in overexpression experiments) or off-target effects (in RNAi experiments).

Reply:

In this manuscript, we showed that ANGPTL8 overexpression inhibited the TNF α -induced NF- κ B activation (revised Fig. 9e), and the knockdown (Fig. 2b,d) or knockout (Fig. 3c,d) of ANGPTL8 showed the opposite effects. To overcome the off-target effects, we used three shRNAs which targeted different regions of the cDNA of ANGPTL8 (the targeting regions are shown below), the results suggested similar inhibitory effects on TNF α -induced NF- κ B

activation (Fig. 2b,d). Furthermore, as a negative control experiment, we confirmed that all three ANGPTL8 shRNAs did not affect IFN γ -induced IRF1 activation, another signaling pathway involved in immunity which separates from TNF α -induced signaling (Fig. 2c). We further confirmed these results in knockdown cells by using the CRISPR-Cas9 system (Fig. 3). In overexpression experiments, each group received equal amount of transfection reagent and plasmids to ensure the experiment groups and the control group having similar stress effects, and we clarify this in the *Methods* section. All these results indicate the specific effects of ANGPTL8 on TNF α -induced NF- κ B activation.

The red regions indicate the target regions of shRNAs for ANGPTL8

As the reviewer suggested, we also performed new experiments to investigate the physiological relevance of ANGPTL8 *in vivo*. First, we challenged mice with LPS. Under such acute inflammatory stress, we measured the protein level of Angptl8 during the acute phase (0-1 hour) and the resolution phase (1-6 hour) among different organs. As reported in the literature¹ and demonstrated by our newly performed qPCR (revised supplementary Fig. 10a), the expression level of Angptl8 is relatively high in the liver and brown adipose tissue (BAT), and low in the spleen, lung and kidney. Interestingly, in tissues with high Angptl8 expression levels, upon LPS challenge, the rates of upregulation and downregulation of *Tnfa* transcription during the acute phase and the resolution phase, respectively, were fast; in contrast, they were relatively slow in tissues with low Angptl8 expression (revised Fig. 10a

and supplementary Fig. 10). These results suggest that for tissues that are sensitive to inflammatory stress, a larger amount of “brake” molecules, such as ANGPTL8, may be demanded.

Second, we detected the interaction among Angptl8, Ikky and p62 in the mouse liver. Under LPS induction, the expression of Angptl8 was increased, while Ikky expression was decreased (revised Fig. 10b,c), however, the interaction between Ikky/p62 and Angptl8 was enhanced (revised Fig. 10d). These results are consistent with *in vitro* observations.

Third, we measured the circulating ANGPTL8 level in the blood of patients with systemic inflammatory response syndrome. In addition to a health control group, two patient groups were selected. One group includes patients with positive detection of procalcitonin (PCT; > 0.5 µg/L) which is a biomarker for early diagnosis of sepsis¹; the other group includes patients with positive detection of endotoxin (> 0.1 EU/mL), which is an important microbiological assessment for gram-negative bacteria-mediated inflammation². Compared with the healthy subjects, the circulating ANGPTL8 level was dramatically increased in both groups of patients (revised Fig. 10e).

Collectively, our new results indicated that ANGPTL8 can be induced by inflammatory stimuli in mice and human, and may play roles in shutting down acute inflammatory response. To our knowledge, this is the first report that associates ANGPTL8 with acute inflammation in clinical samples. Future experiments using *Angptl8* knockout or transgenic mice to reveal the mechanisms underlying its physiological and pathological roles in inflammatory diseases will be extremely helpful. We discuss these in page 18-19 of the revised manuscript.

1. Wacker C, Prkno A, Brunkhorst FM, Schlattmann P. Procalcitonin as a diagnostic marker for sepsis: a systematic review and meta-analysis. *The Lancet Infectious diseases* **13**, 426-

435 (2013).

2. Hurley JC, Guidet B, Offenstadt G, Maury E. Endotoxemia and mortality prediction in ICU and other settings: underlying risk and co-detection of gram negative bacteremia are confounders. *Crit Care* **16**, R148 (2012).

Specific comments:

Fig. 1D shows ANGPTL8 expression in kidney HEK293 cells and lung A549 cells. This is somewhat a surprising finding since ANGPTL8 is not expressed in kidney or lung or in HEK293 and A549 cells according to the protein atlas database. What are the controls that ANGPTL8 is being detected? It is not clear if the experiments are from a single study or replicated in individual experiments.

Reply :

The observation that “ANGPTL8 expression in HEK293 and A549 cell lines, and ANGPTL8 expression can be induced by TNF α ” is reproducible. Data obtained from another two independent experiments performed in these two cell lines are shown below.

As the reviewer suggested, we checked the protein atlas database (<http://www.proteinatlas.org/>) for ANGPTL8 expression level in different cell lines, the

related results listed in the database are appended below. Compared to highly expressed ANGPTL8 in HepG2 cells, the expression levels of ANGPTL8 are usually from mild to not detectable in most cell lines in normal conditions.

Thus, we performed new experiments to compare the ANGPTL8 expression level in the HepG2, HEK293T and A549 cells side by side. HepG2 cell lysate was used as the positive control for ANGPTL8 (detected by WBs). Consistent with the protein atlas database, the level of ANGPTL8 in basal conditions was low in A549 and HEK293T cells, and much higher in HepG2 cells; however, all three tested cell lines demonstrated inducible ANGPTL8 level upon $\text{TNF}\alpha$ stimulation (revised supplementary Fig. 1b). The purpose of the original Fig. 1d was to demonstrate that inflammatory stimulation also induces ANGPTL8 level in other cell lines in addition to HepG2 cells. To help readers focusing on the function of ANGPTL8 in hepatic cells, we have moved the original Fig. 1d to the revised supplementary information as supplementary Fig. 1a.

The paper claims that ANGPTL8 regulates NF- κ B at the IKK complex level. This is shown in Figure 4, where overexpression of IKK β and p65 shows no effect in the presence of overexpression of ANGPTL8 (Fig 4B). However, authors do not explain why they do not see activation by IKK β and p65 overexpression when ANGPTL8 is knocked down (Fig 4B).

Reply:

As the reviewer pointed out, these experiments were not clearly described in the previous manuscript. The same principle and strategy has been applied in signal transduction studies as described in two representative references^{1,2}.

TNF α /IL-1 β -mediated NF- κ B activation includes three major steps: 1) adaptors such as TRAF2/6- or RIP1- mediated recruitment of IKK complex to TNFR; 2) activation of IKK β ; 3) activation of NF- κ B. In the present study, we co-transfected cells with expression vector containing different signaling molecules such as TRAF2/6, or RIP1, or IKK β , or p65, together with a vector overexpressing or knocking down of ANGPTL8 and an NF- κ B luciferase reporter vector, to identify the potential ANGPTL8 affecting molecule(s) in NF- κ B signaling (Fig. 4a,b). In principle, for the cells overexpressing individual signaling molecules such as TRAF2/6, or RIP1, or IKK β , or p65, the pairs showing no difference between the vector control and the ANGPTL8 overexpression/knockdown vector are downstream the molecule(s) affected by ANGPTL8; whereas those molecule(s) showing difference are upstream the site(s) affected by ANGPTL8.

Overexpression/knockdown of ANGPTL8 induced changes of NF- κ B activation under TRAF2/6 or RIP1 overexpression, but not under overexpression of IKK β and p65 (Fig. 4a,b). These data enable us to hypothesize that IKK γ may be a potential ANGPTL8 affecting molecule, because overexpression of TRAF2/6 or RIP1 requires IKK γ for NF- κ B activation,

while overexpression of IKK β or p65 activates NF- κ B independent of IKK γ . As the reviewer suggested, we have revised these descriptions at the *Results* section (page 8).

1. Seth RB, et al. Identification and characterization of MAVS, a mitochondrial antiviral signaling protein that activates NF-kappaB and IRF 3. *Cell* **122**, 669-682 (2005).
2. Cui J, et al. NLRP4 negatively regulates type I interferon signaling by targeting the kinase TBK1 for degradation via the ubiquitin ligase DTX4. *Nature immunology* **13**, 387-395 (2012).

Western blot for RIP1 in Fig 4D should be replaced with better quality image.

Reply:

As the reviewer suggested, we have replaced this western blot with a better quality image (revised Fig. 4d).

Authors claim that ANGPTL8 has a specific effect on the degradation of IKKg as shown in Fig5 A. However, this seems not to repeat in experiment in Fig 7A (Vec control). Authors claim that the degradation of IKKg effect is specific for HA-p62. However, in Fig 7D HA-Tollip also seems to have an effect in IKKg degradation. Quantitation of the Western blots might clarify this effects.

Reply:

As the reviewer suggested, we quantified the western blots according to three independently repeated experiments (shown in revised supplementary Fig. 5 and 6b). Consistent with our previous conclusion, overexpression of ANGPTL8 resulted in significant degradation of IKK γ (quantitative results shown in revised supplementary Fig. 5). To better present the data, in addition to the long-exposure blot for the Flag-IKK γ shown in the original Fig. 7a, a shorter exposure result of the same blot (Flag-IKK γ) was included in revised Fig. 7a.

We agree with the reviewer that from Fig. 7d and the quantified results shown in revised supplementary Fig. 6b, HA-Tollip also shows effects on IKK γ degradation, just like HA-p62. However, the Tollip-related IKK γ degradation was independent of ANGPTL8. Accordingly, we clarified these points in the revised manuscript.

Reviewer #2 (Remarks to the Author):

In this manuscript, Zhang et al. showed that ANGPTL8 and p62/SQSTM1 negatively regulate NF- κ B signaling through autophagic degradation of IKK γ . The paper is well written and the data well presented, but contained some issues to solve before publication in Nature Communications.

We appreciate the reviewer for his/her encouraging comments.

Comments

1. In Figure 6A, to indicate the autophagy-dependent IKK γ degradation, the authors used pharmacological inhibitors such as 3-methyladenine and chloroquine. But, those reagents have side effects other than inhibition of autophagy. The authors should carry out the experiments with Atg-knockout cells (e.g., Atg5, Atg7 and FIP200-knockout cells).

Reply :

We thank the reviewer for this great suggestion. As the reviewer suggested, we performed new experiments using ATG5/7 knockdown cells to detect the effects of ANGPTL8-mediated IKK γ degradation. The results showed that knockdown of ATG5/7 dramatically inhibited the effects of ANGPTL8 on IKK γ degradation, which are consistent with the results

we obtained by using autophagy inhibitors. This piece of new data is shown as revised Fig. 6b in the revised manuscript. Unfortunately, we were unable to obtain any positive clone of ATG5/7 knockout cells using the CRISPR-Cas9 system.

2. *In Figure 6B, the authors should use cells that stably express GFP-LC3 at low level since LC3 is an aggregate-prone protein (Kuma et al., Autophagy. 2007 Jul-Aug;3(4):323-8.).*

Reply :

As the reviewer suggested, we re-performed the experiments using a stably GFP-LC3 expression HepG2 cell line which prevents the unspecific aggregation of LC3. The new representative images and quantification results are shown in revised Fig. 6c, d. The recommended reference is also cited in the revised manuscript.

3. *In Figure 9, the authors identified a region of ANGPTL8, which is needed for its self-oligomerization and the interaction with IKK γ . Using an ANGPTL8 mutant defective IKK γ -interaction, the authors should examine whether the mutant lacks an ability to repress NF- κ B activation in response to TNF α -stimulation.*

Reply:

As the reviewer suggested, we performed additional reporter assay by using an ANGPTL8 Δ 26-70 mutant, and found that this ANGPTL8 mutant, which had impaired interaction ability with IKK γ , could not inhibit the TNF α -induced NF- κ B activation. This new piece of data is shown in revised Fig. 9e.

4. *The authors should test whether autophagic degradation of IKK γ is impaired by expression of a coiled-coil domain mutant of ANGPTL8.*

Reply:

As the reviewer suggested, we studied the ability of the coiled-coil domain of ANGPTL8 (CC, 77-193 aa) and a coiled-coil domain deleted mutant ANGPTL8 Δ CC on the degradation of IKK γ . The results suggested that ANGPTL8 mediates IKK γ degradation independent of its CC domain. This new piece of data is shown in revised supplementary Fig. 7.

5. *The authors may investigate a necessity of LC3-interacting region (LIR) of p62/SQSTM1 in the IKK γ degradation.*

Reply :

As the reviewer suggested, we investigated the involvement of LC3-interacting region (LIR) in the p62-mediated IKK γ degradation by constructing a LIR-deleted p62 mutant (p62 Δ LIR), and found that p62 Δ LIR shows impaired capability in mediating the IKK γ degradation compared with p62. This new piece of data is shown in revised supplementary Fig. 6c.

6. *In IP-IB analysis shown in Figure 4D, the blot for RIP1 was not so clear. Replace it with the representative blot.*

Reply :

As the reviewer suggested, we have replaced the blot for RIP1 with a better quality one.

Reviewer #3 (Remarks to the Author):

The manuscript from Huang and colleagues describes the role of a protein involved in lipid metabolism Angiopoietin-like protein 8 (ANGPTL8) in the regulation of the autophagic degradation of IKK γ under TNF α activation. The authors show that TNF α induces the expression of ANGPTL8 which reduces the activation of NF κ B in TRAF2/TRAF6/RIP1-dependent pathway, suggesting that ANGPTL8 targets the IKK complex. Then they show the ANGPTL8-mediated autophagic degradation of IKK γ using overexpression and knockdown of ANGPTL8. The authors suggest that this occurs through homo- and hetero-dimerization of ANGPTL8 with IKK γ and p62. The study includes a detailed characterization of the mechanism by which ANGPTL8 interacts with IKK γ and p62.

It's an interesting topic and the molecular mechanisms proposed are novel, although the physiological relevance and impact is not addressed. In general the experiments are robust and interpretation appropriate, however, some issues should be addressed before the study could be considered for publication:

We thank the reviewer for his/her comment that “It's an interesting topic and the molecular mechanisms proposed are novel”. We agree with the reviewer that the physiological relevance and impact of ANGPTL8 is not addressed in the previous manuscript. In the revised manuscript, we performed new experiments to address this issue as follows. First, we challenged mice with LPS, and the results suggest that ANGPTL8 may involve in the shutdown of inflammation, especially in some tissues that are sensitive to inflammation. Second, we demonstrated that consistent with our *in vitro* results, the interaction among

IKK γ , p62 and ANGPTL8 *in vivo* exists. Third, we found the expression of circulating ANGPTL8 was dramatically increased in patients with systemic inflammatory response syndrome. All these new data are provided in revised Fig. 10. Collectively, these results suggested ANGPTL8 can be induced by inflammatory stimuli *in vivo*, and may play a role in the inhibition of acute inflammation. Please see *Results* section at page 14-15 of the revised manuscript and also our response to the reviewer #1 for more details.

1) Fig5C; the authors claim only IKK γ is a target for ANGPTL8 mediated autophagy, but levels of IKK α also seem to decrease after ANGPTL8 knockdown.

Reply:

We thank the comment of the reviewer, we quantified the relative expression levels of IKK $\alpha/\beta/\gamma$ based on three independently repeated experiments (see below). The results suggested that knockdown of ANGPTL8 inhibited the proteolysis of IKK γ but not IKK α/β . We replaced the western blot for IKK α with a more representative image according to the quantification results (revised Fig. 5c).

2) Fig6; stable cell lines expressing GFP-LC3 should be used since transient transfection of GFP-LC3 can lead to formation of autophagy-independent dots due to high overexpression (Kuma et al., 2007).

Reply :

We appreciate this great suggestion which was also raised by the reviewer 2. We have repeated the experiments using a stable cell line expressing GFP-LC3 (revised Fig. 6c,e).

3) Fig6C; how has co-localization dots % has been measured? Pearson's and Mander's coefficient should be used (Bolte and Cordelieres, 2006).

Reply :

As the reviewer suggested, we used the plug-in *JACoP* of *Image J* as Bolte et al. described to re-calculate the co-localization rate which is evaluated by Pearson's correlation coefficient and Manders' overlap coefficient¹. Consistent with our previous conclusion, the results indicated that overexpression of ANGPTL8 enhanced the Pearson's and manders' coefficient between IKK γ -Cherry and GFP-LC3 dots significantly, while knockout of ANGPTL8 markedly impaired these coefficients (revised Fig. 6d,f). The related *Methods* section was revised (page 22-23), and the recommended reference is cited in the revised manuscript.

1. Bolte S, Cordelieres FP. A guided tour into subcellular colocalization analysis in light microscopy. *Journal of microscopy* **224**, 213-232 (2006).

4) Fig6D; a clear redistribution of IKK γ from a diffuse localisation to vesicles appears under TNF α stimulation, that does not occur in the absence of ANGPTL8. Is this TNF-induced oligomerisation of IKK γ upon TNF-signalling? This observation supports the data of the authors but is not discussed.

Reply :

We thank the reviewer for this great advice. IKK γ is an important scaffold protein in the TNF α -induced signal pathway. According to literature, TNF α stimulation induces the recruitment of IKK γ into punctate structures, which is essential for NF- κ B activation¹. We cited the reference at page 10 in the *Results* section, and discuss this at page 19 of the revised manuscript as the reviewer suggested.

1. Tarantino N, et al. TNF and IL-1 exhibit distinct ubiquitin requirements for inducing NEMO-IKK supramolecular structures. *The Journal of cell biology* **204**, 231-245 (2014).

Again, we appreciate the reviewers for their insightful comments, which helped us to improve this work. We hope the present manuscript is more suitable for publication.

REVIEWERS' COMMENTS:

Reviewer #1 (Remarks to the Author):

I have one minor comment:

Ref 48 (Yi et al., 2013) should be removed since the paper was retracted. The name "Betatrophin" should be removed from the text since ANGPTL8 is not a betatrophin (reason for retraction)

Reviewer #2 (Remarks to the Author):

The revised manuscript was greatly improved. Just one thing: To prove the inhibition of autophagy by knock-down of Atg5 or Atg7, the authors should show the immunoblot of LC3 in Figure 6B.

Point-by-point response letter

Re: Manuscript NCOMMS-17-01143A entitled “ANGPTL8 negatively regulates NF- κ B activation by facilitating selective autophagic degradation of IKK γ ” .

Reponses to reviewers' comments

Reviewer #1 (Remarks to the Author):

I have one minor comment:

Ref 48 (Yi et al., 2013) should be removed since the paper was retracted. The name "Betatrophin" should be removed from the text since ANGPTL8 is not a betatrophin (reason for retraction)

Reply:

It has been revised.

Reviewer #2 (Remarks to the Author):

The revised manuscript was greatly improved. Just one thing: To prove the inhibition of autophagy by knock-down of Atg5 or Atg7, the authors should show the immunoblot of LC3 in Figure 6B.

Reply:

As the reviewer suggested, we showed the immunoblots of LC3 in the control and ATG5/7-RNAi cell lines in revised Figure 6B. We further confirmed these results under rapamycin treatment (revised Supplementary Figure 5). These results suggested that knockdown of ATG5 or ATG7 dramatically blocked autophagy both in the basal level and under rapamycin stimulation.